

# Aerosol-induced changes in the vertical structure of precipitation: a perspective of TRMM precipitation radar

Jianping Guo[1], Huan Liu[1], Zhanqing Li[24*], Daniel Rosenfeld[3], Mengjiao Jiang[4], Weixin Xu[5], Jonathan H. Jiang[6], Jing He[1], Dandan Chen[1], Min Min[7], and Panmao Zhai[1]

[1]State Key Laboratory of Severe Weather, Chinese Academy of Meteorological Sciences, Beijing 100081, China

[2]Department of Atmospheric and Oceanic Sciences & Earth System Science Interdisciplinary Center, University of Maryland, College Park, Maryland 20740, USA

[3]The Hebrew University of Jerusalem, Jerusalem 91904, Israel

[4]College of Global Change and Earth System Sciences, Beijing Normal University, China

[5]Department of Atmospheric Sciences, Colorado State University, Fort Collins, Colorado 80523, USA

[6]Jet Propulsion Laboratory, California Institute of Technology, Pasadena, California 91109, USA

[7]NationalSatelliteMeteorological Center, China Meteorological Administration, Beijing 100081, China

*Correspondence to*: Zhanqing Li (zli@atmos.umd.edu)



## Abstract

This study investigates aerosol effects on precipitation over the Pearl River Delta region of China using six years of ground-based $PM_{10}$ and satellite-based (TRMM) precipitation data. In general, rain rate tends to be lower under polluted conditions than under clean conditions. Radar reflectivity of the top 1% increases as the atmosphere becomes slightly polluted ($PM10 < 38$ μg/m$^3$), except for shallow convection. The aerosol-precipitation data pairs are further limited to local- or meso-scale

precipitation systems. Results show that significant changes in precipitation vertical structure are possibly induced by aerosol, and this potential aerosol effect is regime dependent. The 30 dBZ radar echo top height is elevated by 18.7% (2.7%) for convective (stratiform) precipitation under severe polluted conditions ($PM_{10} > 83$ μg/m$^3$) compared to clean conditions ($PM_{10} < 31$ μg/m$^3$), indicative of a possible aerosol invigoration effect. In contrast, the 30 dBZ radar echo top height

of shallow convection are almost identical between pristine and polluted conditions. Impacts of meteorological factors are further studied on both echo top and reflectivity center of gravity, including vertical velocity, vertical wind shear, convection available potential energy, and vertically integrated moisture flux divergence. The possible invigoration effect on convective precipitation seems dependent on wind shear, in good agreement with previous simulations.

Overall, the observed dependence of precipitation vertical structure on ground-based $PM_{10}$ supports the aerosol invigoration hypothesis and adds a new insight into the nature of the complex interactions between aerosol and various precipitation regimes.

## 1 Introduction

Despite many challenges and uncertainties remaining, there have been increasing evidences/phenomena showing the impact of aerosol on weather and climate including extreme events like severe thunderstorms, as recently reviewed by Li et al. (2017). Convective invigoration

has been suggested in ample studies that both the height (Williams et al., 2002; Andreae et al., 2004; Koren et al., 2005; Jiang et al., 2008; Rosenfeld et al., 2008; Li et al., 2011; van den Heever et al., 2011; Fan et al., 2013) and fraction (Fan et al., 2013; Yan et al. 2014) of deep convective clouds increase with aerosol loading, thereby leading to stronger storms in polluted environments. At the same time, the inhibition of light precipitation by aerosols has also been reported in different

regions of the world (Kaufman and Fraser, 1997; Rosenfeld and Lensky, 1998; Rosenfeld and Givati, 2006; Wang et al., 2011; Guo et al., 2014). The invigoration theory was recently



generalized by Fan et al. (2018) that can also occur for shallower water clouds under extreme clean conditions under which ultra-fine mode aerosol particles may be nucleated to release latent heat to fuel cloud development. While we have come a long way in understanding the mechanisms behind various observation-based findings, the impact of aerosol on precipitation remain a daunting task

(Tao et al, 2012). Failure in fully understanding and accounting for these effects may not only undermine our understanding of the earth's climate and its changes (IPCC, 2013), but also impair the accuracy of rainfall forecast by a numerical weather model (Jiang et al., 2017).

The specific effects of aerosols on precipitation are strongly influenced and confounded by atmospheric dynamic and thermodynamic conditions, such as updraft strength (Koren et al., 2012;

Tao et al., 2012; Guo et al., 2016), wind shear (Fan et al., 2009), and atmospheric instability (Gordon, 1994; Khain et al., 2004). By serving as cloud condensation nuclei (CCN), high aerosol concentration leads to more but smaller cloud droplets that consume the available water vapor more efficiently (Koren et al., 2010). Consequently, aerosols can indirectly modify the vertical profiles of hydrometeors and cloud phases, which can, in turn, alter the dynamics and

thermodynamics of a precipitating cloud system through latent heat release (Heiblum et al., 2012). Relationships between aerosols and precipitation also vary significantly on seasonal and spatial scales (Huang et al., 2009a,b,c). It has been a great challenge to single out the aerosol effects, largely due to various processes influencing precipitation, radiation, and even the state of the atmosphere that are induced by aerosols.

The three-dimensional (3D) structure of radar echoes are known to represent the vertical structure of precipitation, which is determined by a combination of dynamic, thermodynamic, and microphysical processes occurring in precipitation systems (Yuter and Houze, 1995; Heiblum et al., 2012). Any systematic changes in the vertical structure of precipitation as aerosol loading changes may provide some insights into the mechanism underlying the aerosol-cloud-precipitation

interaction. The deployment of the cloud profiling radar onboard CloudSat has indeed led to new insights into cloud and precipitation microphysical processes (e.g., Nakajima et al., 2010; Suzuki et al., 2010; Chen et al., 2016). Studies examining aerosol effects on precipitation systems using satellite observations (e.g., Rosenfeld, 2000; Niu and Li, 2012; Peng et al., 2016) are often limited to column-integrated aerosol optical depth (AOD) and cloud top properties.



Given the dominant effects of cloud dynamics on synoptic-scale precipitation systems, only precipitation events occurring on local scale are examined in detail in the following sections. This consideration is largely due to local- or meso-scale precipitating clouds, including the thermal convection, cumulus, and stratocumulus clouds, are less dependent on large scale dynamic

conditions and more susceptible to aerosol pollution (Fan et al., 2007; Lee et al., 2012; Guo et al., 2017). The goal of this study is to investigate the influence of aerosols on the vertical structure of different precipitation types on local scale by examining a large amount of collocated measurements from the precipitation radar (PR) on board the Tropical Rainfall Measuring Mission (TRMM) and ground-based in-situ aerosol measurements made in the Pearl River Delta (PRD)

region of southern China. We will examine differences in the vertical structure of precipitation between clean and dirty atmospheric environments to determine whether they are consistent with some previously proposed mechanisms governing aerosol invigoration or suppression of precipitation.

The rest of this paper proceeds as follows. We first describe the data and methods in section 2,

including definitions of several parameters describing vertical precipitation radar echoes. In section 3, we examine any dependence of the vertical structure of precipitation on aerosols in terms of radar reflectivity and analyze how they change with meteorological factors as well. Finally, we summarize the major findings in section 4.

**2 Data and methods**

2.1 Study area

The region of interest is the PRD region (113E°-115°E, 22°N-24°N, bounded by red rectangles in Figure 1), which includes many populated cities such as Guangzhou, Shenzhen, Zhuhai, Hong Kong, Macau, Dongguan, Zhongshan, and Foshan. The PRD has a humid

subtropical climate, which is strongly influenced by Asian monsoon circulations and tropical cyclones originating in the western Pacific Ocean (Ding, 1994). The PRD region has been experiencing rapid economic development in recent years. As a result, high concentrations of aerosol particles associated with human activities (e.g., increasing fossil fuel combustion due to industrialization) have been frequently documented (Deng et al., 2008; Guo et al., 2009).

Observations have shown that precipitation and the frequency of lightning have been enhanced in



recent years in southern China, as atmospheric pollution worsened in the region (Wang et al., 2011; Yang and Li, 2014).

## 2.2 Data

The datasets used in this study are listed in Table 1 and are briefly described here. Precipitation data from the TRMM PR and aerosol data collected at the surface from 1 January 2007 to 31 December 2012 are used in this study, unless noted otherwise. Aerosol loading information retrieved by space-borne sensors is limited to cloud-free conditions, leading to a lack of coincident and collocated measurements of aerosols and precipitation. Particulate matter (PM) with an aerodynamic diameter less than 10 µm ($PM_{10}$) measured at surface is thus used as a proxy of aerosol loading. Meteorological variables are taken from the European Centre for Medium-Range Weather Forecasts (ECMWF) reanalysis data (Uppala et al., 2008).

### 2.2.1 TRMM PR and 3B42 data

Precipitation 3D structures are obtained from the TRMM PR (Kummerow et al., 1998). The TRMM PR 2A23, 2A25, and 3B42 products (version 7) are used in this study. Precipitation is classified into convective and stratiform types as provided in the 2A23 product. In order to better characterize the local-scale precipitation (detailed in section 3.2) and its association with aerosols, a third precipitation type, namely shallow precipitation type, is included in this study. Shallow precipitation is simply defined as the shallow isolated echo category in 2A23. The 2A25 product provides vertical profiles of radar reflectivity and rain rates at a vertical resolution of ~250 m ranging from the near-surface level to 20 km altitude. 2A25 data have a horizontal resolution of 4-5 km depending on the satellite orbit height and the PR off-nadir view angle. All pixels that do not exceed the radar reflectivity threshold of 15 dBZ (the minimum detectable reflectivity factor for the TRMM PR) are omitted (Kummerow et al., 1998). To make sure that a pixel in question contains a reliable precipitation event, the following criteria are used in this study: (1) the attenuation-corrected reflectivity (Z) must be equal to or greater than 15 dBZ. Namely, it is equivalent to a rain rate (R) of 0.2 mm/h. The Z-R conversion is calculated using the equation of $Z = 200R1.6$, which applies for continental stratiform precipitation (Marshall and Palmer, 1948); and (2) there must be at least four consecutive levels with $Z \geq 15$ dBZ. The TRMM 3B42 version 7 product merges precipitation radar and microwave rainfall estimates with infrared-based



precipitation estimates from multiple satellites, as well as measurements from rain gauges (Huffman et al., 2007). The estimates are gridded at a 0.25°x0.25° spatial resolution over the global belt between 50°N and 50°S and have a three-hour temporal resolution.

### 2.2.2 Ground-based $PM_{10}$ measurements

Given the difficulties in obtaining large-scale CCN concentration information, we have to resort to any CCN proxy such as satellite-derived AOD and the aerosol index, or ground-based PM measurements. Previous studies (e.g., Koren et al., 2005, 2012; Jiang et al., 2008; Huang et al., 2009b) have shown that there are sound correlations between satellite retrievals of AOD, and cloud and precipitation properties. Such correlations are susceptible to various uncertainties arising

from cloud contamination and the dependence of AOD on certain atmospheric components like water vapor (e.g., Li et al., 2009; Boucher and Quaas, 2012). Moderate Resolution Imaging Spectroradiometer (MODIS) AOD products are available for less than 30% of the time over the PRD region (Wang et al., 2015). Very large uncertainties arise when using AOD as a proxy for CCN (Andreae, 2009). These uncertainties can be reduced by applying the method proposed by

Liu and Li (2014). However, the most serious problem in using AOD as a proxy for CCN lies in the fact that AOD is only measurable under cloud-free conditions and is subject to various retrieval errors, as critically reviewed by Li et al. (2009).

The ability of a particle to nucleate a cloud droplet depends on its size and its chemical composition. The aerosol index is defined as the product of AOD and the Angström exponent, and

is a good proxy to use to quantify CCN due to its ability to weight AOD measurements towards the fine mode (Nakajima et al., 2001; Andreae, 2009). A limitation of using the aerosol index is that retrievals are restricted to over oceans because of the large uncertainties in Angström exponent retrievals over land (Levy et al.,2010).

Given the above problems, we choose to use the rich dataset of ground-based $PM_{10}$

observations in the PRD region, which are available from 1 January 2007 to 31 December 2012. While it would be better to use $PM_1$ and $PM_{2.5}$ (Seinfeld and Pandis, 1998), much less such data are available for matching with TRMM data during the period selected for this study. Using a recent year of coincident $PM_{2.5}$ and $PM_{10}$ measurements at the region studied here, we found a good correlation (PM2.5/PM10> 0.7) between the two variables (Figure 1b). Because this study is

concerned with establishment of the contemporaneous association of radar echo reflectivity with




various aerosol loadings, using PM$_{10}$ (available under all sky conditions) as a proxy for CCN is sufficient for our needs. Vertical profiles of aerosols and clouds over the PRD region obtained from the Cloud-Aerosol Lidar and Infrared Pathfinder Satellite Observations mission show that aerosol particles are generally well-mixed in the boundary layer (Wang et al., 2015). PM$_{10}$ data

can then indicate major aerosol episodes over the relatively small domain in the PRD region (~200km x 200km). According to Anderson et al. (2003), the variability in aerosol properties at such a spatial scale is not very large.

### 2.2.3 Reanalysis data

Due to the meteorological factors influencing simultaneously aerosol concentration and

precipitation, it will be more feasible if the investigation of the co-variation of aerosol and precipitation is considered under the same meteorological conditions based on ECMWF ERA-Interim reanalysis data (Uppala et al., 2008). Meteorological parameters used include vertical velocity ($\omega$), specific humidity, the "u" component of wind (U), the "v" component of wind (V), and convective available potential energy (CAPE). These data are available four times a day, with

a horizontal resolution of 0.125°×0.125° at pressure levels equal to 1000, 975, 950, 925, 900, 875, 850, 825, 800, 775, 750, 700, 650, 600, 550, 500, 450, and 400 hPa. The relationship between aerosols and precipitation structure can be established when the dataset is sorted out according to meteorological variables (Koren et al., 2012).

## 2.3 Methodology

### 2.3.1 Stratification of precipitation using PM$_{10}$ measurements

Precipitation types, including shallow, stratiform, and convective precipitation, are directly derived from the TRMM 2A23 product. We only consider cases with simultaneously available measurements of both PM$_{10}$ and precipitation. This study only attempts to differentiate data

corresponding to the lowest and highest tercile of PM$_{10}$ concentration to denote the cleanest and dirtiest conditions, rather than any among the subtler differences that would require more precise information of aerosol. The PM$_{10}$ dataset is divided into three bins with each bin containing an equal number of samples. Table S1 summarizes the range of PM$_{10}$ values defined for each bin as well as the mean value of PM$_{10}$ in each bin for shallow, stratiform, and convective precipitation

types. Data are divided into three groups to make sure that the daily mean PM$_{10}$ concentration





exceeds the national air quality standard for the polluted case (75 µg/m3). The first (lowest) bin represents clean conditions and the third (highest) bin represents polluted conditions. As such, the clean conditions correspond to average $PM_{10}$ concentration of 27.5 µg/m$^3$, 23.6 µg/m$^3$ and 24.4 µg/m$^3$ for shallow, stratiform and convective precipitation regimes, respectively, while polluted

ones correspond to 120.6 µg/m$^3$, 99.9 µg/m$^3$ and 97.6 µg/m$^3$. On average, clean conditions for all precipitation types are defined when the daily mean $PM_{10}$ is <38 µg/m$^3$ and polluted conditions are defined when the daily mean $PM_{10}$ is >102 µg/m$^3$ (Table 2).

It also creates a sufficient contrast between clean and polluted groups while retaining good sampling statistics (Koren et al., 2012). Table 2 summarizes the total number of profiles and the

frequency of occurrence (in %, relative to the total number of profiles) of profiles in the clean and polluted categories for each precipitation type. To further examine aerosol influences on convective precipitation, this precipitation type is divided into three groups based on hourly R: light (R < 10 mm/h), moderate (10 ⩽ R < 20 mm/h), and heavy (R ⩾ 20 mm/h). The percentage of convective precipitation samples in each precipitation intensity category is summarized in Table

S2.

*2.3.2 Meteorological variables*

Previous aerosol-precipitation interaction studies have suggested that atmospheric dynamic conditions and moisture fluxes are among the most important meteorological variables contributing to changes in cloud properties and associated precipitation (Koren et al., 2010;

Medeiros and Stevens, 2011; Jiang et al., 2011). To better isolate the aerosol effect, we need to determine the relative contributions of the following four meteorological factors to the variability in precipitation: ω, CAPE, vertical wind shear between 1000 hPa and 700 hPa, and vertically integrated moisture flux divergence (MFD) from 1000 hPa to 400hPa.

CAPE is a measure for the amount of moist static energy for initiation of convection, and acts

as an effective indicator of atmospheric instability, which has been shown to be closely associated with the initiation of precipitation (Dai et al., 1999). For a fixed atmospheric condition, wind shear can dictate whether aerosols suppress or enhance convective strength, depending on the atmospheric moisture and stability (Fan et al., 2009). MFD, another major factor in the formation of precipitation, determines the complex spatial variability of precipitation through the transport

of water vapor (Khain et al., 2008).





The definition of MFD in units of g/(cm²s) is:

$$MFD = \nabla_P \cdot \left( \frac{\overrightarrow{V_H}q}{g} \right) = \frac{\partial}{\partial x}\left( \frac{\overrightarrow{V_H}q}{g} \right) + \frac{\partial}{\partial y}\left( \frac{\overrightarrow{V_H}q}{g} \right) \tag{1}$$

$$\overrightarrow{V_H} = \vec{U} + \vec{V} \tag{2}$$

where $\overrightarrow{V_H}$ represents the horizontal wind vector, $\vec{U}$ and $\vec{V}$ represent the U and V components of wind

in units of m/s, q represents specific humidity in units of g/kg, P represents pressure in units of hPa,

and g represents the acceleration due to gravity. MFD was calculated at 18 standard pressure levels:

1000, 975, 950, 925, 900, 875, 850, 825, 800, 775, 750, 700, 650, 600, 550, 500, 450, and 400 hPa.

A negative MFD means convergence of water vapor and a positive MFD indicates divergence of

water vapor.

*2.3.3 Normalized contoured frequency by altitude diagram*

The contoured frequency by altitude diagram (CFAD) displays the height-resolved occurrence

frequencies of Z (Yuter and Houze, 1995). The CFAD ignores variation in t and x and retains only

variation in Z as observed by PR. The CFAD helps in examining the evolution of an ensemble of

small-scale variabilities in differential reflectivity. The occurrence frequency of the jth Z at the ith

level, CFAD$_{ij}$, is written as

$$CFAD_{ij} = \frac{\int_{H_i}^{H_i+\Delta H} \int_{Z_j}^{Z_j+\Delta Z} \frac{\partial^2 N(H,Z)}{\partial H \partial Z} dZ dH}{\Delta Z \Delta H \int_{-\infty}^{\infty} \frac{dN(Z)}{dZ} dZ} \tag{3}$$

where $N(H,Z)$ is the frequency distribution function defined as the number of observations of $Z$ in

the range of $Z$ to $Z+\triangle Z$ at a height above ground ranging from $H$ to $H+\triangle H$. The index $i$ goes

from 1 to 80 (in intervals of 0.25 km) and the index $j$ goes from $1$ to $60$ (in intervals of 1 dBZ).

There may be times when there are few occurrences of $Z$ in a particular range of $H$. To overcome

this problem, an improved statistical technique known as the normalized contoured frequency by

altitude (NCFAD) has been widely used (e.g., Fu et al., 2003). The improvement comes from




normalizing the frequency at each altitude level to the total number of points at all levels, which

is expressed as

$$NCFAD_{ij} = \frac{\int_{H_i}^{H_i+\Delta H} \int_{Z_j}^{Z_j+\Delta Z} \frac{\partial^2 N(H,Z)}{\partial H \partial Z} dZ dH}{\Delta Z \Delta H \int_0^{Htop} \int_{-\infty}^{\infty} \frac{\partial^2 N(H,Z)}{\partial H \partial Z} dZ dH} \qquad (4)$$

TRMM PR observed $Z$ profiles matched to ground-based $PM_{10}$ measurements are used to construct

the NCFAD. To highlight the aerosol effect on the vertically-evolving process of precipitation,

NCFAD plots (polluted minus clean conditions) are constructed.

*2.3.4 Reflectivity center of gravity*

The bulk precipitation system parameter called the reflectivity center of gravity (ZCOG;
*Rosenfeld and Ulbrich*, 2003) is used to represent the vertically-weighted reflectivity distribution.

The ZCOG can cancel out any systematic reflectivity biases throughout the vertical profile

(Rosenfeld and Ulbrich, 2003). The ZCOG indicates the height where the total Z mass tends to

concentrate and is highly sensitive to precipitation microphysical and dynamical processes (Koren

et al., 2009; Heiblum et al., 2012). It is defined as

$$ZCOG = \frac{\sum_i Z_i H_i}{\sum_i Z_i} \qquad (5)$$

where $Z$ is the measured radar reflectivity in dBZ, $H$ is the height above ground in km, and $i$ is an

index from 1 to 80, representing different levels in the atmosphere. A larger magnitude of ZCOG

means that the precipitation system has developed to a higher level in the atmosphere, indicating

stronger convection.

**3 Results and discussion**

3.1 Regional aerosol features

$PM_{2.5}$ began to be measured as of 2013, largely due to the "January 2013" severe haze event

shrouded over the whole eastern China. China central government decided to make great efforts



in attempt to address the increasingly serious air quality issues across the board, including setting up the $PM_{2.5}$ criteria, among others. Therefore, $PM_{2.5}$ measurements during the period of January 2007 - December 2012 do not exist. It is still an efficient alternative way to use the yearly averaged PM data during the period of November 2013-October 2014 to characterize the regional aerosol features in the PRD region. Figure 1a presents the spatial distribution of mean $PM_{10}$ concentrations collected in the PRD region from November 2013 to October 2014. Nearly 60% of the measurement sites are characterized by high PM10 concentrations (>70 μg/m$^3$). This value (70 μg/m$^3$) is the World Health Organization (WHO) interim target 1 annual mean level, which is associated with about a 15% higher long-term mortality risk relative to the WHO air quality guideline level of 20 μg/m$^3$ (WHO, 2006).

Figure 1b shows the ratio of annual mean $PM_{2.5}$ to annual mean $PM_{10}$. Most megacities (e.g., Guangzhou and Shengzhen) are characterized by a high ratio of $PM_{2.5}$ to $PM_{10}$ (> 0.7). This suggests that fine PM, which is mostly generated by anthropogenic activities such as daily power generation and industrial production, dominates aerosol pollution in this area. This region is an ideal testbed to probe the aerosol impact on 3D precipitation structures.

3.2 Discrimination between synoptic-and local-scale precipitating systems

Generally speaking, synoptic-scale precipitation involves frontal passages or low-pressure systems, as compared with local-scale precipitation characterized by thermal-driven convective clouds fed by the boundary layer air (aerosol). Our recent study (Guo et al., 2017) indicates that local-scale precipitation events are more closely linked to aerosol relative to synoptic-scale precipitation. In order to make sure that only precipitating system more susceptible to the boundary layer aerosol were considered, all the satellite scenes with synoptic precipitation were excluded. For any given day, ground-based aerosol observations have to collocate with precipitation measurements from TRMM in attempt to obtain a valid data pair. As such, the total number of collocated samples reached up to 255 for local-scale precipitation events, whereas 194 for synoptic scale precipitation events. The local-scale precipitation event was determined based on the weather charts, where daily averaged wind field at 850 hPa was plotted along with geopotential height at 500hPa. Note that the discrimination was manually performed through visual interpretation of the weather plot for each day with valid precipitation (>0.1 mm/day) over PRD, owing to the extreme complexities in discriminating the weather systems for local- and synoptic-scale precipitations.





Figure 2 illustrates two typical weather plots, corresponding to synoptic- and local-scale precipitation events, respectively. On 26 June 2008 (Figure 2a), PRD lies at the bottom of the weak low pressure at 500 hPa level. At 850 hPa level, there is a weak cyclone on the left-forward side of PRD, where a south-western to north-eastern low-level jet stream overpasses at the same time,

leading to strong water vapors advected over PRD from South China Sea. More importantly, the wind shear observed at 850 hPa is most favorable for the formation and evolution of precipitation. Overall, the weather patterns at both 500 hPa and 850 hPa help the onset and development of large-scale convection, so this precipitation event occurred over PRD can be thought of as a typical synoptic-scale precipitation event. In contrast, PRD is largely controlled by the subtropical high-

pressure areas, in combination with the anti-cyclone systems at low levels on 2 July 2008, as shown in Figure 2b. Therefore, this precipitation can be attributed to local thermal convection with high certainty.

3.3 The contemporaneous association of radar reflectivity and aerosol

In this section, the possible aerosol effect on precipitation at a regional scale is investigated without consideration of precipitation type. Precipitation enhancement or inhibition by aerosols is examined by comparing R under polluted and clean atmospheric conditions. Daily mean R is first calculated over the PRD region. Figure 3 shows the geographical and frequency distributions of differences in R. Differences are calculated as R under polluted conditions minus R under clean

conditions. Caution must be exercised in the interpretation of the TRMM 3B42 precipitation product because a droplet size distribution affected by the presence of pollution (producing more and smaller drops) would lead to a different Z-R relation, which also depends on the microphysical, dynamical and topographical context of the precipitating clouds (Rosenfeld and Ulbrich, 2003). This may be what is happening in Figure 3a, which shows a few grid boxes where precipitation

enhancement occurs during polluted conditions. The frequency distribution of differences in R (Figure 3b) further shows that negative differences in R can be seen over roughly 30% of the study area under polluted conditions compared with clean conditions. In other words, ~70% of the study area has an increased R when aerosol loading increases. These statistical results appear to support in some way the notion of precipitation enhancement by increases in aerosol pollution, but at this



stage, the effect of meteorological factors described in section 2.3.2 on precipitation cannot be excluded.

A few recent studies (Koren et al., 2014; Wang et al., 2015) have shown that less developed cloud and precipitation are very sensitive to aerosol when the atmosphere transitions from pristine to slightly polluted conditions. Therefore, focusing on the initial stage of precipitation evolves with aerosol, the lowest tercile (Table 2) of $PM_{10}$ concentrations matched with their corresponding Z values are plotted. Figure 4 shows the average occurrence frequency (OF) in each $Z/PM_{10}$ concentration bin for shallow, stratiform, and convective precipitation types. There is little systematic change in Z with aerosol loading for all precipitation types (solid black lines). The top 1% OFs for convective precipitation, however, has an increasing trend in Z as the aerosol loading changes from pristine to slightly polluted, i.e., $PM_{10}$ concentration varies from 0 to roughly 38 $\mu g/m^3$, as shown in dashed black line of Figure4c. The trend stabilizes at relatively high $PM_{10}$ concentrations. Given that meteorological variables are not correlated with $PM_{10}$ (cf. FiguresS1-S2 in Supporting Information), aerosols are assumed to be able to invigorate precipitating convective clouds with larger reflectivity when the aerosol loading is relatively low, which is the same as in the stratiform precipitation case to some extent. For stratiform precipitation, as aerosol loading continuously increases, the top 1% OF for each bin of radar reflectivity goes up sharply then levels off. In other words, the aerosol invigoration effect is observed for stratiform precipitation, which largely occurs as the atmosphere becomes slightly polluted ($PM_{10}$<38 $\mu g/m^3$). By contrast, there is no distinct variation in reflectivity with aerosol loading for shallow precipitation.

3.4 The vertical structure of precipitation associated with aerosols

The vertical structure of precipitation (in the form of radar reflectivity) to some extent represents the convective intensity and precipitation microphysics of a precipitation system (Zipser and Lutz, 1994; Yuan et al., 2011). Due to the intrinsic dependence of R on Z (cf. Figures S3 in Supporting Information), changes in the vertical structure of Z as a function of aerosol concentration, if any, can indicate aerosol effects on convective intensity and precipitation formation. Differences in Z profiles between polluted and clean conditions for shallow, stratiform, and convective precipitation types are examined next.





Figure 5 shows differences in vertical profiles of the frequency of occurrence of Z between polluted and clean cases for shallow, stratiform, and convective precipitation types. The most striking finding is the well-defined features of positive and negative differences dominant in different parts of the plotting domain. Had aerosols had no effect, we would see mixed colors

without such distinct patterns. As explained below, not only are the patterns well defined, but also are robust statistics well behaved, which is consistent with the well-established theories of aerosol-cloud interactions (e.g., Rosenfeld et al., 2008; Li et al., 2011; Tao et al., 2012).

As expected, convective precipitation is more vertically developed than shallow and stratiform precipitation. For shallow precipitation (Figure5a), there is an echo at the maximum

height of 4-5 km, where the frequency of occurrence of $\triangle Z$ (polluted - clean) is negative, a likely sign that aerosols might have suppressed precipitation at these heights. Moreover, for Z greater than 24 dBZ, negative $\triangle Z$ values dominate, suggesting the inhibiting effect of aerosols. Z values less than 24 dBZ are more frequent under polluted conditions below 3 km. This is because the cases that would have been more intense (from right to left in Figure5a), lead to a reduced

frequency on the right (blue) and an enhanced frequency on the left (red). In general, the pattern for stratiform precipitation (Figure5b) is similar to that of shallow precipitation, except for its development to relatively higher altitudes.

Convective precipitation has a totally different NCFAD pattern (Figure 5c). Above 5 km and for radar echoes stronger than 45 dBZ which are mostly mixed-phase or ice processes, the

overwhelming warm colors denote that precipitation echoes in the presence of heavy aerosols tend to be lifted to higher altitude than those in the low aerosols. Below the freezing level as the reflectivity is less than 45 dBZ, the color is virtually all blue, meaning that precipitation is weaker under polluted conditions than clean ones. The reversal behavior of radar echo intensity around the freezing level for stratiform and convective clouds can hardly be explained by any

meteorological factors unless they are correlated with $PM_{10}$, which seems not the case according to Figures S1-S2 in the Supporting Information part of this paper. A more plausible, but unnecessarily the sole explanation roots on aerosol microphysical effects, which lead to the invigoration of cloud and precipitation above the freezing level at the expense of lower levels (Rosenfeld et al., 2008; Li et al.,2011). Aerosol microphysical and radiative effects on precipitation

usually interact and sometimes cancel each other out, leading to either invigoration or suppression



(Rosenfeld, 2000; Zhang et al., 2007; Rosenfeld et al., 2008), with both effects being found from such long-term measurements as the ARM (*Li et al.*, 2011). Aerosols have an invigorative or suppressive effect depending on various factors, such as wind shear, humidity, cloud water amount, precipitation intensity (Fan et al., 2009; Li et al., 2011; Guo et al., 2014).

Figure 6 shows NCFADs of ΔZ for the different precipitation intensities associated with convective precipitation. Positive difference for the radar echoes above the freezing level (roughly 5km) in the presence of aerosols can be seen for convective precipitation regardless of precipitation intensity. Interestingly, negative difference dominates below about 5 km level for light precipitation, but the magnitude is much larger for moderate to heavy precipitation than that for

light precipitation. For radar precipitation echoes<30 dBZ, NCFAD patterns are similar in all categories of precipitation intensity. Shaded areas corresponding to positive values are seen from 9-15 km where radar precipitation echoes are mostly less than 30 dBZ. In a convective system, this is closely linked to internal microphysical properties and lightning. The enhancement of 30 dBz reflectivity above the freezing level is often associated with larger ice particles and more

super-cooled liquid water contents (Zipser, 1994). Hence, the differences shown in Figure 6 could be indicative of an aerosol-invigorated convective echo occurring above the freezing level. Differences observed in the internal structure may also reflect differences in updraft velocities, and thus heating rates. Parts of the atmosphere with updraft velocities less than a certain threshold value tend to have less ice particles and ice-ice collisions in the mixed-phase region above the

freezing level (Zipser, 1994). This further complicates the potential aerosol invigorative effect. Resolving such an ambiguity would require much more detailed in-situ measurements from a dedicated field experiment, which is beyond the scope of this study.

Another way of ascribing internal Z differences in convective echoes to differences between polluted and clean conditions is to consider the maximum height of the 30 dBZ echo. Figure7

shows that the 30 dBZ echo heights of convective (stratiform) precipitation are on average elevated (decreased) from 4.4 km (4.3 km) under clean condition to 5.6 km (3.9 km) when $PM_{10}$ concentration reaches the highest third bin of 97.6μg/m$^3$ (99.9μg/m$^3$), or an increase of 27.3% (-10.3%), while no any increase or decrease can be seen in 30 dBZ radar echo height for shallow precipitation. In other words, the frequencies of occurrence of stratiform (convection) precipitation

under polluted conditions are generally lower (higher) than those under clean conditions for all 30 dBZ maximum heights, as indicated in Figure7b (Figure 7c). These generally agree with the results



shown in Figures 4-5. Overall, the difference is statistically significant in terms of average height between the polluted and clean cases, except for shallow precipitation (Table 3). Combining Figures 4, 5 and 7 with Table 3, an invigoration (suppression) effect for convection (stratiform) precipitation types can be observed, which may be partly due to the aerosol radiative,

microphysical or combined effect on the vertical development of various precipitation systems. But at this stage, such influence cannot be attributed to aerosols alone. Although all of above analyses are restricted to local-scale precipitation, the effect of meteorology still has to be taken into account as well, which will be discussed in the following section.

3.5 The aerosol-meteorology-precipitation dilemma

The radar echo top height is one parameter that has been considered as the key to describing the vertical structure of a population of radar echoes (Houze and Cheng, 1977; López, 1977). However, some detailed internal variations in the vertical structure of Z cannot be explained by analyzing echo top height alone. The ZCOG is representative of the internal structure of Z to some

degree. As a result, both echo top height and the ZCOG are used to examine the vertical structure of convective echoes in association with aerosol pollution. The aerosol indirect effect may not entirely account for the systematically different NCFADs observed under polluted versus clean atmospheric conditions. It is well known that aerosols and precipitation systems are simultaneously influenced by the meteorology, which is also dubbed as a buffered system due to the nonlinear

dependence between them (Steven and Feingold, 2009). The observed association of aerosols with precipitation vertical structure in above sections should then be further analyzed as a function of relevant meteorological factors.

Mean echo top heights and ZCOGs under clean and polluted conditions as functions of $\omega$, vertical wind shear, CAPE, and MFD, and for the different precipitation regimes are shown in

Figure 8. To make the statistics more robust, each bin in a particular panel is equally-spaced. The standard deviations of echo top height and ZCOG are calculated for each bin as well. As shown in Figures 8i-k, both the convective precipitation echo top height and ZCOG under polluted atmospheric conditions are located at higher altitudes than those under clean atmospheric conditions, except for those under high wind shear (Figure 8l). This trend is generally opposite to

what is seen for shallow and stratiform precipitation, which further corroborates the notion of an




aerosol invigoration effect on convective precipitation and a suppression effect on shallow and stratiform precipitation types as shown in Figure 5. More interesting is that unstable atmospheric, weak vertical wind shear and relatively humid conditions favor more convective precipitation invigoration, as indicated by the relatively large magnitudes in Figure 8i-k, which is highly
consistent with previous observational and modeling studies (Khain et al., 2008; Fan et al., 2009; Gonçalves et al., 2015). In particular, both convective precipitation echo top heights and its ZCOGs tend to develop to higher (lower) altitudes in the presence of aerosols when the vertical wind shear is smaller (larger), as opposed to the responses of echo top heights and ZCOGs for the same wind shear conditions for shallow precipitation (Figure 8c). This is consistent with previous
findings reported by Fan et al. (2009) who pointed out that increasing the aerosol loading suppresses convection under strong wind shear conditions and invigorates convection under weak wind shear conditions.

A closer look at Figure 8 reveals that stratiform and convective precipitation types have larger differences in terms of both echo top height and ZCOG, as compared with shallow precipitation.
In addition, the differences in echo top height can be easily detected for both stratiform and convective precipitation regimes, unlike the observed differences in ZCOG under polluted and clean conditions. No obvious positive difference is seen in shallow precipitation, except for a subtle elevated echo top height and ZCOG observed under high CAPE conditions.

When the atmosphere becomes thermodynamically unstable (greater ω in Figure 8a and larger
CAPE values in Figure 8c), the negative difference in the echo top height of shallow precipitation between polluted and clean conditions becomes more evident, likely indicative of aerosol suppression effect in this case. This effect is facilitated by the less vertically integrated MFD (Figure 8b). This could be due to the fact that in the dry environment characteristic of the study area, the inhibitive effect of aerosols on shallow precipitation easily stands out in the absence of a
thermodynamically stable atmosphere.

**4 Conclusions**



Many studies have been reported concerning the impact of aerosol on the bulk properties of cloud and precipitation using meteorological data and satellite retrievals chiefly from passive sensors. This study establishes some contemporaneous relationships between radar echo and aerosol over the Pearl River Delta (PRD) region using TRMM precipitation radar (PR) reflectivity

(Z) profiles and precipitation rate estimates, in combination with ground-based $PM_{10}$ measurements. In particular, the association of the changes in vertical structure of precipitation with aerosols is investigated in attempt to figure out the possible aerosol effect on precipitation for shallow, stratiform and convective regimes respectively, which are all restricted to local-scale precipitating systems.

Concerning the mean joint frequency of occurrence (OF) for each $PM_{10}$/Z bin, there are almost no systematic changes in Z as $PM_{10}$ concentrations change, irrespective of precipitation type. Z increases as aerosol loading increases for stratiform and convective precipitation types in the top 1 % of OFs as the atmosphere transitions from pristine to slightly polluted conditions. There is no distinct variation in reflectivity with aerosol loading for shallow precipitation. The indicated

aerosol effects, as evaluated by contrasts in the normalized contoured frequency by altitude diagram (NCFAD) of ΔZ, are shown to systematically discriminate between different vertical structures associated with shallow, stratiform, and convective precipitation types. Overall, convective precipitation tends to develop to much higher altitudes compared with shallow and stratiform precipitation. Below the freezing level (~5 km), the occurrence of reflectivity>45 dBZ

was enhanced at the expense of the probability of lower reflectivity. This crossover point decreased to ~20 dBZ near 9 km. This is consistent with convective echo enhancement due to the aerosol effect above the freezing level. Enhancement of the 30dBz reflectivity above the freezing level is often associated with presumably larger super-cooled liquid water and ice particles. This leads to larger and more reflective hydrometeors and to possible invigoration by aerosols. Due to the

fundamental role of convective precipitation in the hydrological cycle, the aerosol microphysical effect on convective precipitation has been further examined with regard to light, moderate, and heavy convective precipitation types, a measure of precipitation intensity. The NCFADs of ΔZ were similar, irrespective of precipitation intensity.

Under polluted conditions, a statistically significant increase in the mean height of 30 dBZ

radar echo top for convective precipitation is seen under polluted conditions, as opposed to the



slightly increased mean height for the 30 dBZ radar echo top of stratiform precipitation, suggesting that aerosols can enhance the radar echo of convective precipitation to some extent.

The relationship between aerosols and bulk precipitation parameters such as radar echo top and ZCOG, stratified by specific ω, vertical wind shear, CAPE, and MFD, were also examined in

an attempt to disentangle aerosol impacts on the vertical structure of precipitation from meteorology. There is no systematic signal of aerosol or meteorology on the development of shallow and stratiform precipitation. In contrast, under certain meteorological conditions, apparent difference in the response of echo top and ZCOG for stratiform and convective precipitation types to the aerosols can be seen. But under some extreme conditions, the observed difference in

response was confounded by the meteorology, partly due to the fact that meteorology affects both aerosol and precipitation systems. For instance, weak vertical wind shear and relatively humid conditions typically come with the possible aerosol-induced invigoration of convective precipitation observed in this study, in good agreement with previous model simulation (e.g., Khain et al., 2008; Fan et al., 2009).The results presented here provide some sound but not

unequivocal evidence of the possible impact of aerosol on the vertical structures of three different types of precipitation regimes, due to the common inherent aerosol-meteorology-precipitation dilemma. The relationships between changes in TRMM PR reflectivity and aerosol perturbations are statistically significant and consistent with the existing theories, but they may be subject to different interpretations concerning the underlying physical processes. Confirming or negating any

causes with confidence would require a much more detailed knowledge of the cloud processes than the satellite observation used here, and should be further aided by model simulations of aerosol-cloud-precipitation interactions.

**Acknowledgements**

The authors would like to acknowledge NASA for making the TRMM precipitation radar satellite datasets publicly accessible, as well as the NASA-sponsored Jet Propulsion Laboratory, California Institute of Technology for support. The CMORPH precipitation data can be accessed and downloaded from the China Meteorological Data Sharing Service System (http://cdc.cma.gov.cn/home.do). The $PM_{10}$, $SO_2$, and $NO_2$ data were obtained from the

Guangzhou Environmental Protection Bureau (http://www.gzepb.gov.cn/comm/pm25.asp). All the original datasets and code needed to reproduce the results shown in this paper are available



upon request. This study was supported by the Ministry of Science and Technology of China (Grant 2017YFC1501401), the National Natural Science Foundation of China (Grants 91544217, 41771399 and 41471301) and the Chinese Academy of Meteorological Sciences (Grant 2017Z005).




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



**Tables**

**Table 1**. Specifications from TRMM PR retrieved precipitation, National Atmospheric Environment Observation Network (NAEON) in situ measured $PM_{10}$, and ECMWF reanalysis meteorological data used
5    in this study for the period of 1 January 2007 to 31 December 2012. Criteria for selecting data for further comprehensive analysis are provided in the footnote.

| Source | | Data | Horizontal resolution | Vertical resolution | Temporal resolution |
|---|---|---|---|---|---|
| TRMM | 2A23 | Rain type | 5.0 km | - | daily* |
| | 2A25 | Reflectivity | 5.0 km | 0.25 km | daily* |
| | | Rain rate | 5.0 km | 0.25 km | daily* |
| | 3B42 | Precipitation | 0.25°×0.25° | - | Three-hourly |
| NAEON | | $PM_{10}$ | - | - | Hourly |
| ECMWF | | Vertical velocity | 0.125°×0.125° | - | Six-hourly |
| | | Convective available potential energy | 0.125°×0.125° | - | Six-hourly |
| | | U component of wind | 0.125°×0.125° | - | Six-hourly |
| | | V component of wind | 0.125°×0.125° | - | Six-hourly |
| | | Specific humidity | 0.125°×0.125° | - | Six-hourly |

| Criteria | (1) TRMM worked normally; |
|---|---|
| | (2) Precipitation-fall measured by TRMM PR; |
| | (3) There must be at least four consecutive levels with Z≥15 dBZ for a given profile. |

*calculated from the times of the TRMM PR swath over passing the PRD region.*



**Table 2**. Statistics describing the three precipitation types considered in the study. Occurrence frequencies for each precipitation type (relative to the total number of precipitation profiles) are given in percent. The $PM_{10}$ thresholds discriminating between clean (bottom 1/3) and polluted (top 1/3) conditions and their corresponding numbers of precipitation profiles and percentages (relative to the total number of precipitation profiles for that precipitation type) are also listed. Data are from TRMM PR retrievals made over the PRD region.

| Precipitation type | # of profiles | % | Clean ($PM_{10}$) | | | Polluted ($PM_{10}$) | | |
|---|---|---|---|---|---|---|---|---|
| | | | Threshold ($\mu g/m^3$) | # of profiles | % | Threshold ($\mu g/m^3$) | # of profiles | % |
| Shallow | 846 | 10.4 | ⩽40 | 586 | 69.3 | ⩾77 | 212 | 25.1 |
| Stratiform | 5360 | 66.0 | ⩽35 | 1998 | 37.3 | ⩾60 | 797 | 14.9 |
| Convective | 1912 | 23.6 | ⩽34 | 572 | 29.9 | ⩾59 | 930 | 48.6 |



**Table 3.** Statistics describing the mean heights of 30 dBz radar echoes under polluted and clean

| Precipitation type | # of clean samples | # of polluted samples | Ave. height of clean 30 dBz echoes(km) | Ave. height of polluted 30 dBz echoes (km) | Abs.(T) for a=0.05 |
|---|---|---|---|---|---|
| shallow | 172 | 29 | 2.40 | 2.56 | 1.24(×) |
| stratiform | 1089 | 351 | 4.34 | 3.87 | *12.37( √ )* |
| convective | 483 | 816 | 4.36 | 5.63 | *11.29( √ )* |

conditions for different precipitation types. The numbers in bold italics indicate that the difference

between polluted and clean mean 30 dBz heights are statistically significant at the 95% confidence level

5 according to the Student's t test.



**Figures**

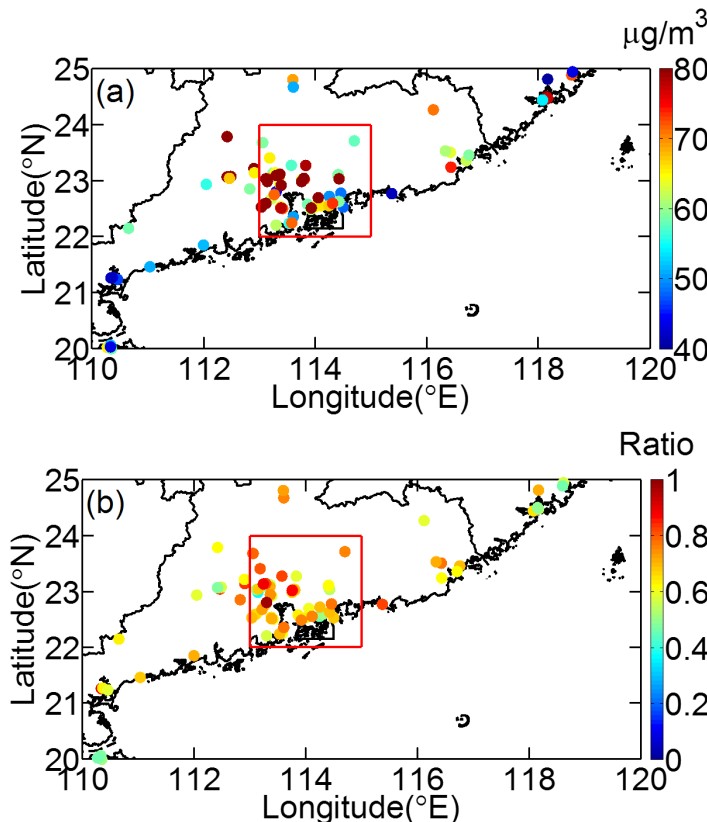

**Figure 1.** Spatial distributions of (a) ground-based measured mean $PM_{10}$ (in μg/m³) and (b) the ratio of mean $PM_{2.5}$ to $PM_{10}$ over the PRD region from November 2013 to October 2014, when

5   data quality checks on the PM data were done. Note that $PM_{2.5}$ began to be measured as of 2013, and the red box outlines the region of study and the dots show the locations of the PM measurement sites.





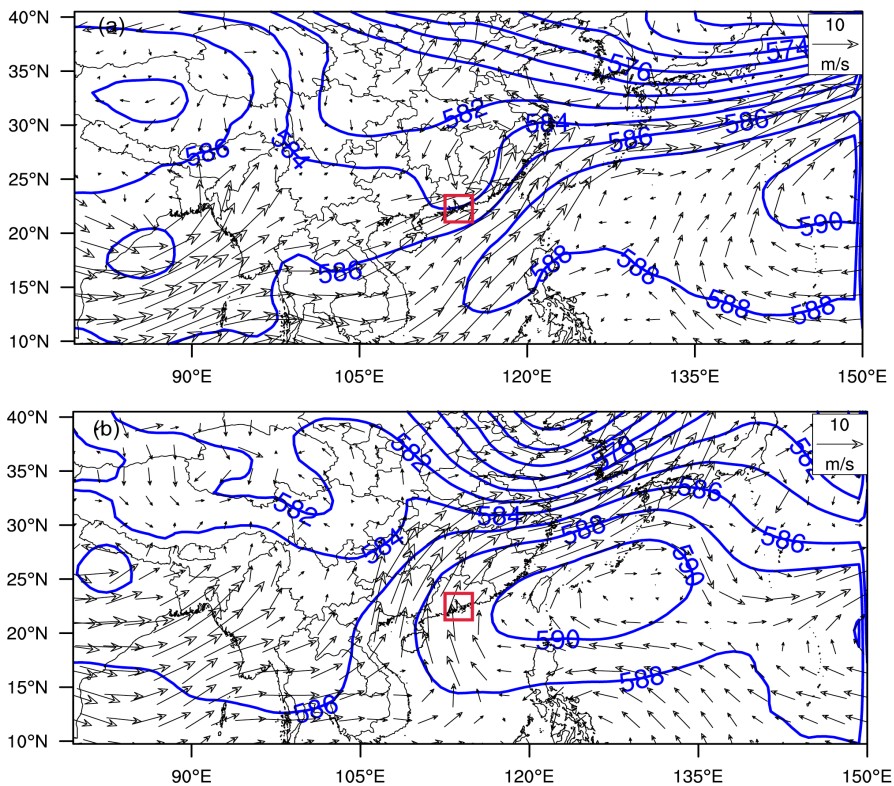

**Figure 2.** Charts showing the wind field at 850 hPa (black arrows, vector), superimposed by geopotential height at 500 hPa (blue lines) averaged on 26 June 2008 (a), and 2 July 2008 (b). All data are from the ECMWF ERA-Interim reanalysis data, and the red rectangle denotes the study area.





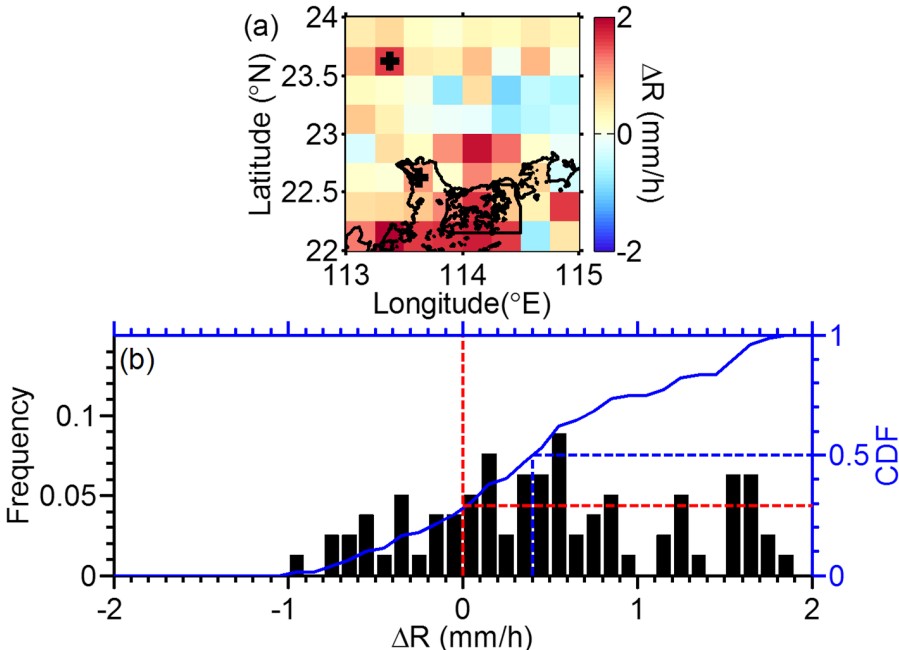

**Figure 3.** (a) Differences in precipitation intensity (polluted minus clean conditions, mm/h) over the PRD region. The black dots show grid boxes for which the difference exceeds the 95% significance level ($p < 0.05$) according to the two-tailed Student's t-test. (b) Histogram showing the occurrence frequency (OF) and its cumulative distribution frequency (CDF) of precipitation intensity differences between polluted and clean conditions. The threshold value used to discriminate between clean and polluted atmospheric conditions corresponds to lowest and highest tercile of the $PM_{10}$ concentration averaged over the PRD region, respectively. The points where blue and red dashed lines cross correspond to cumulative probabilities of 29%.




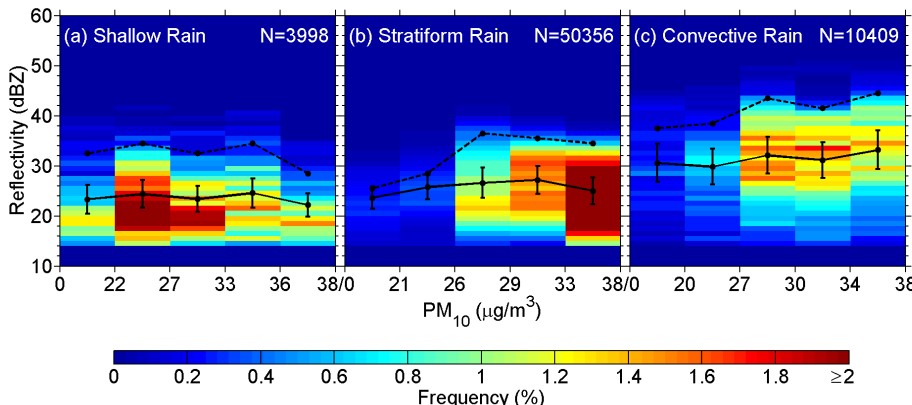

**Figure 4.** Two-dimensional histograms of the mean occurrence frequency of (a) shallow, (b) stratiform, and (c) convective precipitation derived from TRMM 2A23 products for altitudes ranging from 1-5km during the period 2007-2012. Colors indicate the average frequency in each radar reflectivity and $PM_{10}$ concentration bin. The top 1% (mean) with respect to occurrence frequency for each $PM_{10}$ concentration bin is represented by dashed (solid) black lines. The number of profiles, N, used for the calculation of frequency is shown in the upper-right corner of each panel. Note that the lowest tercile of $PM_{10}$ concentration is used here to highlight the aerosol effect on precipitation





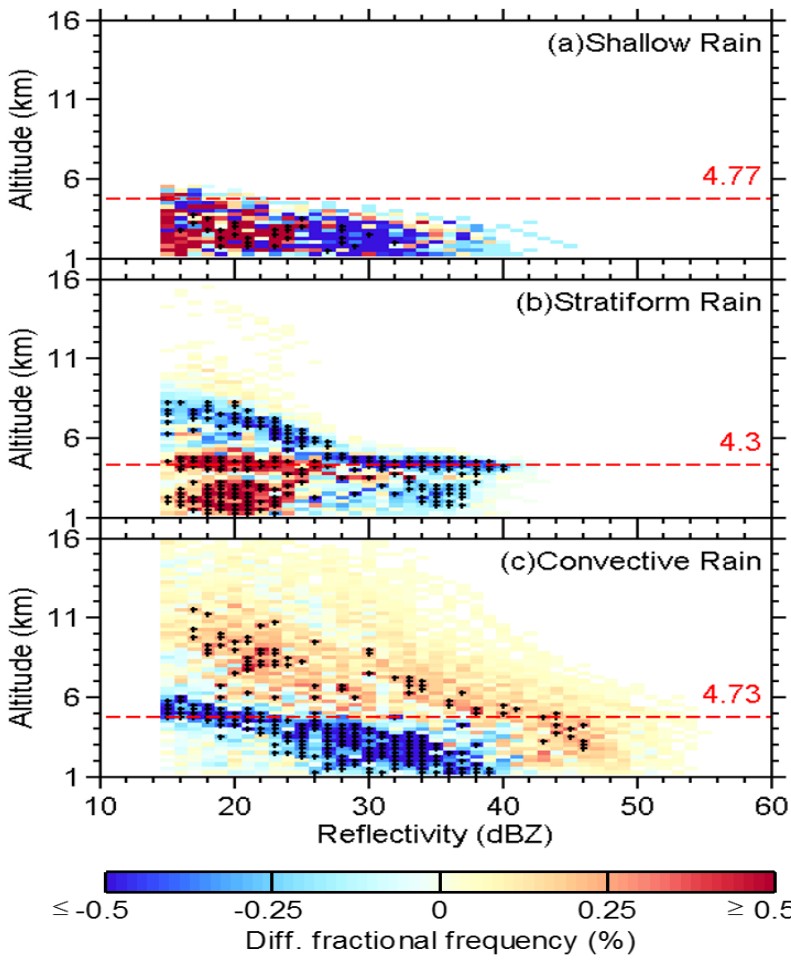

**Figure 5.** NCFAD diagrams of differences in the frequency of occurrence for detected precipitation echoes (polluted minus clean) for (a) shallow precipitation, (b) stratiform precipitation, and (c) convective precipitation types. Data are from TRMM PR retrievals made during 2007-2012. The horizontal red dashed lines show the freezing level and the black crosses mark grid points where the difference exceeds the 95% significance level ($p < 0.05$) according to the Pearson's $\chi$-square test.



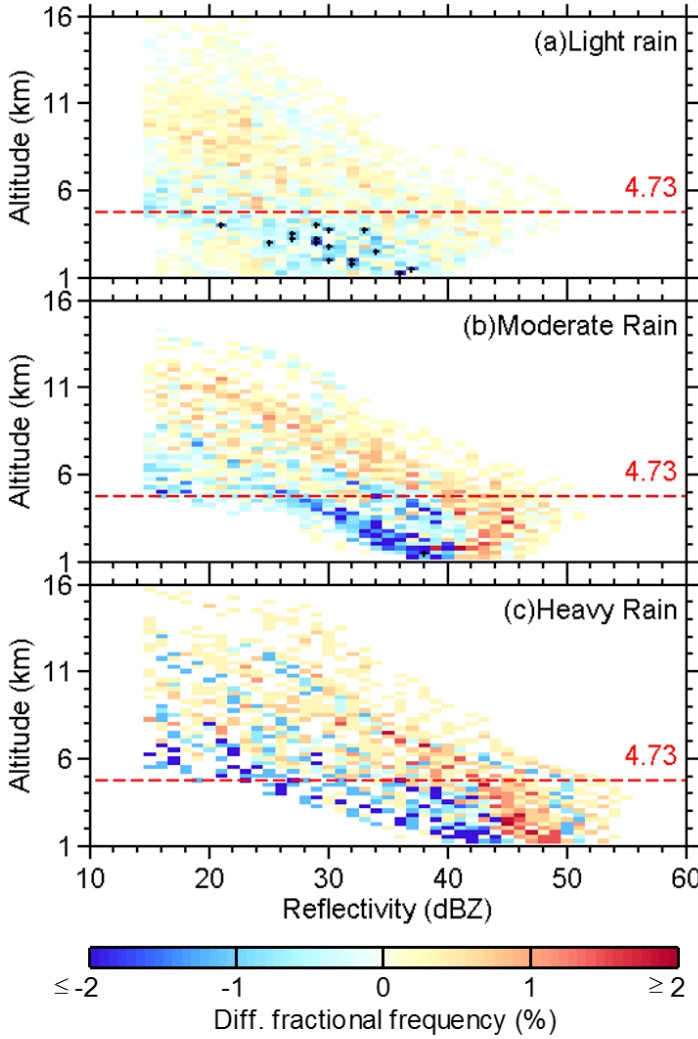

**Figure 6.** NCFAD diagrams of differences in the frequency of occurrence for detected convective precipitation echoes (polluted minus clean) for (a) light precipitation, (b) moderate precipitation, and (c) heavy precipitation. Data are from TRMM PR retrievals made during 2007-2012. The horizontal black dashed lines show the freezing level and the black crosses mark grid points where the difference exceeds the 95% significance level ($p < 0.05$) according to the Pearson's $\chi$-square test.





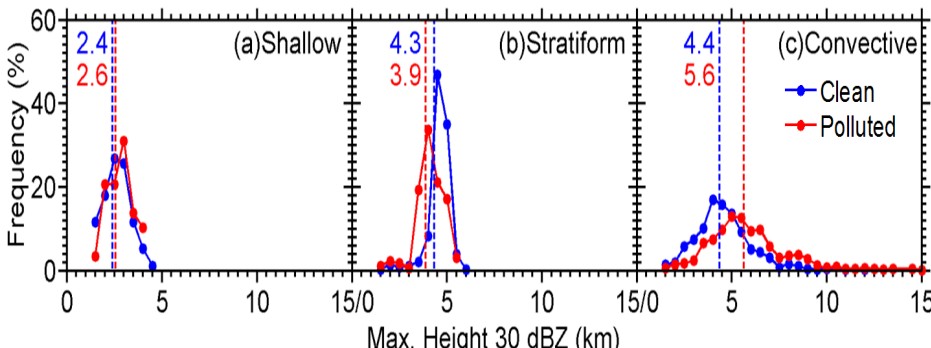

**Figure 7.** Occurrence frequencies (OF) of the maximum height of the 30 dBz radar echo of (a) shallow, (b) stratiform, and (c) convective precipitation. Data are from TRMM PR retrievals made during 2007-2012. Red and blue colors represent polluted and clean cases, respectively. Vertical lines represent average heights of the 30 dBZ radar echoes. Dots on the profile curves indicate vertical levels at which the differences in the distributions of the clean and dirty Z values are statistically significant at the 99% level based upon the Kolmogorov–Smirnov test.

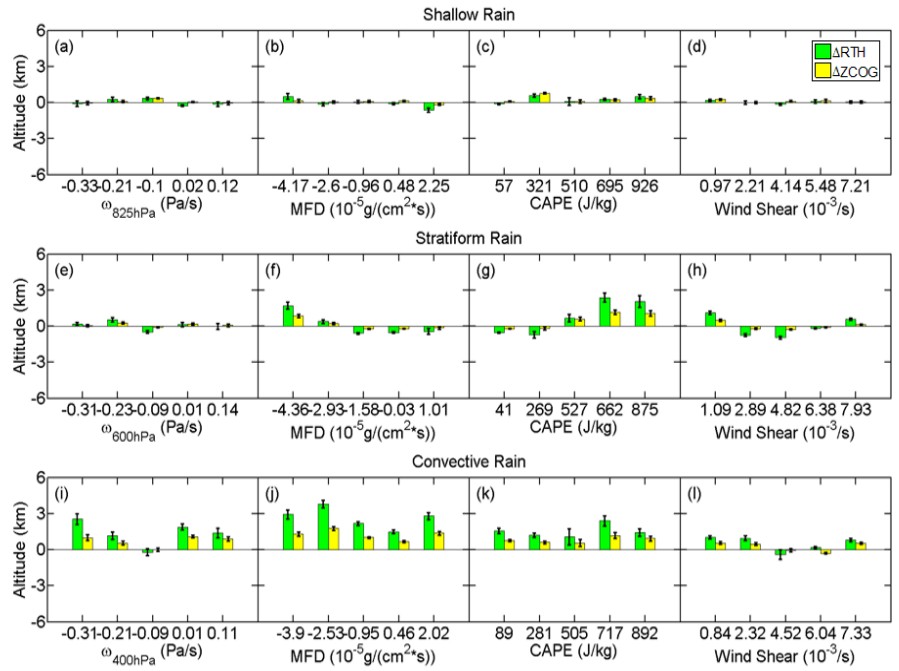



**Figure 8.** The 18 dBz radar echo height differences (ΔRTH, polluted minus clean) and ZCOG difference (ΔZCOG, polluted minus clean) as a function of (a) ω at 600 hPa for shallow precipitation, (b) MFD for shallow precipitation, (c) CAPE for shallow precipitation, (d) vertical wind shear for shallow precipitation, (e) ω at 600 hPa for stratiform precipitation, (f) MFD for stratiform precipitation, (g) CAPE for shallow stratiform precipitation, (h) vertical wind shear for stratiform precipitation, (i) ω at 600 hPa for convective precipitation, (j) MFD for convective precipitation, (k) CAPE for convective precipitation, and (l) vertical wind shear for convective precipitation. Data are from 2007-2012. Note that negative ω refers to upward motion. Red and blue colors represent polluted and clean cases, respectively. The vertical error bars represent one standard deviation. Each bin has an equal number of samples.