# Peer review of "Aerosol-induced changes in the vertical structure of precipitation: a perspective of TRMM precipitation radar"

_Atmospheric Chemistry and Physics, 2018_

## Short Comment (SC1) · 24 May 2018

General comments: The study utilized the TRMM radar reflectivity and PM10 data over the PRD region to investigate the potential impacts of aerosol on precipitation. How to quantify aerosol impacts on precipitation based solely on observations is a tough task since meteorological factors need to be isolated effectively. The study is unique in that it separated precipitation associated with synoptic or mesoscale forcing from those localized precipitation events. Furthermore, other meteorological factors, including vertical wind shear, which is important for convective system development, are also analyzed and described. The finding that aerosol is able to invigorate deep

convections is generally consistent with previous modeling studies. The study is thus a good contribution to this community. Nevertheless, I have some suggestions for the authors to consider.

Specific comments:

1. Although manual identification of synoptic or localized precipitation event is described on 26-28 p11, a few more description might help since it is very subjective. I am also wondering whether localized precipitation is more appropriate than local-scale precipitation.

2. Smaller reflectivity below the freezing level for polluted cases than clean cases (Fig. 5c, P14 Line 10) might be due to the large numbers, but smaller sizes of rain drops within polluted environment.

3. Looks like PM10 (P8, Line 5) is much higher during the periods with occurrence of shallow convection than other two types of precipitation. Any reasons for this? Does this imply heavy pollution tends to inhibit deep convection development sometimes, although it will invigorate deep convection once the negative impacts of aerosols are overcome.

4. Regarding that deep convections sometimes developed from shallow convections, is it possible that the composite will divide one precipitation event into different types. This need to be mentioned somehow.

Minor comments:

1. Why use the vertical wind shear between 1000 and 700 hPa instead over a higher level?

2. P6, L20, delete "use to"

3. there are some other typos. Please double check.

---

## Referee Comment (RC1) · Anonymous Referee #1 · 25 May 2018

General commentsïijŽ The study utilized the TRMM radar reflectivity and PM10 data over the PRD region to investigate the potential impacts of aerosol on precipitation. How to quantify aerosol impacts on precipitation based solely on observations is a tough task since meteorological factors need to be isolated effectively. The study is unique in that it separated precipitation associated with synoptic or mesoscale forcing from those localized precipitation events. Furthermore, other meteorological factors, including vertical wind shear, which is important for convective system development, are also analyzed and described. The finding that aerosol is able to invigorate deep convections is generally consistent with previous modeling studies. The study is thus a good contribution to this community. Nevertheless, I have some suggestions for the

authors to consider.

Specific comments 1. Although manual identification of synoptic or localized precipitation event is described on 26-28 p11, a few more description might help since it is very subjective. I am also wondering whether localized precipitation is more appropriate than local-scale precipitation. 2. Smaller reflectivity below the freezing level for polluted cases than clean cases (Fig. 5c, P14 Line 10) might be due to the large numbers, but smaller sizes of rain drops within polluted environment. 3. Looks like PM10 (P8, Line 5) is much higher during the periods with occurrence of shallow convection than other two types of precipitation. Any reasons for this? Does this imply heavy pollution tends to inhibit deep convection development sometimes, although it will invigorate deep convection once the negative impacts of aerosols are overcome. 4. Regarding that deep convections sometimes developed from shallow convections, is it possible that the composite will divide one precipitation event into different types. This need to be mentioned somehow.

Technical corrections:

1. Why use the vertical wind shear between 1000 and 700 hPa instead over a higher level? 2. P6, L20, delete "use to" and some other typos. Please double check.

Please also note the supplement to this comment:
https://www.atmos-chem-phys-discuss.net/acp-2018-366/acp-2018-366-RC1-supplement.pdf

---

## Short Comment (SC2) · 31 Jul 2018

The paper entitled "Aerosol-induced changes in the vertical structure of precipitation: a perspective of TRMM precipitation radar" by Guo et al. is nicely written and presenting a good quality work on the response of vertical features of precipitation to aerosol in the PRD region of China using TRMM observations. Author used a long data series consisting of 6 years of data and it is really a good for statistical analysis on aerosol-precipitation interaction and there is scope for further extending the work on aerosol-induced changes in inner dynamics of precipitation in future. However, some results are not properly discussed.

[Figure]

1. Page 16 line 4: Regarding the results of "An invigoration (suppression) effect for convection (stratiform) precipitation types can be observed", the authors may consider to add the following discussion in Manuscript:

When the tropical convection systems developed and matured, its precipitating cloud cluster consists as follows: 1) deep precipitating convective towers characterized by vigorous updrafts; 2) stratiform precipitating cloud connected to the deep convection exhibiting weaker, mesoscale vertical motions; 3) non-precipitating thick anvils attached to either the stratiform or convective precipitating areas (Houze 1993; Li and Schumacher, 2011;Yang et al., 2015). Because the samples are mainly restricted to local-scale precipitation (non large-scale precipitation) in the present work, the selected stratiform precipitation samples are mainly connected to the deep convection in tropical PRD regions. High-concentration aerosols over PRD easily tend into these deep convection under strong upward moment with enough water vapor, which will favor the development of convective precipitations, while for vertically weaker motion stratiform precipitation, high-concentration aerosols and non-abundant water vapor are supplied to suppress the development of stratiform precipitation.

Houze, 1993:Cloud Dynamics. International Geophysics Series, Vol. 53, Academic Press, 573 pp. Li, W., and Schumacher, C. (2011). Thick Anvils as Viewed by the TRMM Precipitation Radar. Journal of Climate, 24(6):1718–1735. Yang Yuan-Jian, Da-ren Lu, Yun-Fei Fu, et al., 2015.Spectral Characteristics of Tropical Anvils Obtained by Combining TRMM Precipitation Radar with Visible and Infrared Scanner Data, Pure and Applied Geophysics., 172, (6), 1717-1733 DOI:10.1007/s00024-014-0965-x.

2. Page 16 line 4: Regarding the inferrence of aerosol radiative effect in "which may be partly due to the aerosol radiative", key reference is missing, for example, Liu et al. GRL 2018

Liu Z., Yim S.H.L., Wang C., Lau N.C. (2018). The impact of the aerosol direct radiative forcing on deep convection and air quality in the Pearl River Delta region. Geophysical

Research Letters, 45(9), 4410-4418"

---

## Referee Comment (RC2) · Anonymous Referee #2 · 2 Aug 2018

Review of "Aerosol-induced changes in the vertical structure of precipitation: a perspective of TRMM precipitation radar" by Guo et al.

Taking the Pearl River Delta region of China as the study area, this paper reported the observational evidence of aerosol effect on precipitation using TRMM and PM10 datasets. Different precipitation types or regimes in particular the vertical structure of precipitation are differentiated for more explicit explanation. The findings from the study are convincing, and the technical details are sufficient to support the results. The impacts from other meteorological factors are also discussed in order to single out the aerosol effects on precipitation. The study is at the frontier of this research field, and I

highly recommend the paper is published in ACP at current format. I only have a few minor comments and suggestions for the authors to consider if they are going to revise the paper further for improvements: 1. Page 6 L24: problems -> considerations? 2. Page 7 Line 25: tercile -> terciles? 3. P7L26: dirtiest -> most polluted? 4. P7L27: bins -> terciles? 5. Table 2: would it be possible to add PM10 information in this table as well? Such as Mean and Standard Deviation of PM10 for Clean and Polluted terciles? 6. Page 8 L5-7: will such definitions of clean and polluted conditions match the terciles with each tercile having same number of samples? 7. Figure 1 caption: is PM10 in (a) collected over the same period of Pm2.5? If not, it is better to be specific. 8. Figure 2 reminds me how the seasonal cycles in the datasets are treated. For example, PM10 are collected from different months, did you deseasonalize the datasets before the three terciles are determined? This will help avoid high tercile Pm10 are mostly collected from one season, and low tercile are from another season. It is therefore worthwhile and critical to check whether your samples from each tercile are biased to seasons. 9. P13 L12: Figure4c -> Figure 4(c) 10. PM10 datasets availability: Did PM10 data end in December 2012 and then replaced by PM2.5 in 2013? Why we don't have any PM10 any more after 2012 to extend the study period longer? It is strange they stopped measuring PM10 after they switched to PM2.5 in 2013. 11. Figure 6: did you conduct the similar thing for shallow and stratiform precipitation? 12. Figure 7 and 8: I am not sure I understand them completely. But I do like the discussions associated with them. Great work!

---

## Author Comment (AC2) · 16 Aug 2018

**Response to Short Comment #2 by Dr. Zengyun Hu**

**General comments:**

The paper entitled "Aerosol-induced changes in the vertical structure of precipitation: a perspective of TRMM precipitation radar" by Guo et al. is nicely written and presenting a good quality work on the response of vertical features of precipitation to aerosol in the PRD region of China using TRMM observations. Author used a long data series consisting of 6 years of data and it is really a good for statistical analysis on aerosol-precipitation interaction and there is scope for further extending the work on aerosol-induced changes in inner dynamics of precipitation in future. However, some results are not properly discussed.

***Response: First of all, we appreciate the positive and constructive comments provided by Dr. Zengyun Hu. In response to his comments, we have made relevant revisions to the manuscript. Listed below are our responses and the corresponding changes made to the manuscript according to the suggestions offered by him. Each comment is echoed in normal front, followed by our responses in bold italics.***

**Minor comments:**

1. Page 16 line 4: Regarding the results of "An invigoration (suppression) effect for convection (stratiform) precipitation types can be observed", the authors may consider to add the following discussion in Manuscript:

When the tropical convection systems developed and matured, its precipitating cloud cluster consists as follows: 1) deep precipitating convective towers characterized by vigorous updrafts; 2) stratiform precipitating cloud connected to the deep convection exhibiting weaker, mesoscale vertical motions; 3) non-precipitating thick anvils attached to either the stratiform or convective precipitating areas (Houze 1993; Li and Schumacher, 2011; Yang et al., 2015). Because the samples are mainly restricted to local-scale precipitation (non large-scale precipitation) in the present work, the selected stratiform precipitation samples are mainly connected to the deep convection in tropical PRD regions. High-concentration aerosols over PRD easily enter into these deep convection under strong upward moment with enough water vapor, which will favor the development of convective precipitations, while for vertically weaker motion stratiform precipitation, high-concentration aerosols and non-abundant water vapor are supplied to suppress the development of stratiform precipitation.

References:

Houze, 1993: Cloud Dynamics. International Geophysics Series, Vol. 53, Academic Press, 573 pp.

Li, W., and Schumacher, C. (2011). Thick Anvils as Viewed by the TRMM Precipitation Radar. Journal of Climate, 24(6):1718–1735.

Yang Yuan-Jian, Da-ren Lu, Yun-Fei Fu, et al., 2015.Spectral Characteristics of Tropical Anvils Obtained by Combining TRMM Precipitation Radar with Visible and Infrared Scanner Data, Pure and Applied Geophysics., 172, (6), 1717-1733

DOI:10.1007/s00024-014-0965-x.

*Response: Thanks a lot for a very clear elaboration of the development of a deep convective clouds and its association with aerosols, which alos points out the difficulties in classifying precipitation types. The significant role that vertical motion plays in convective and stratiform clouds has been analyzed in Figure 8, conditioned by the supply of water vapor. Therefore, part of your comments has been incorporated into section 3.1, as follows:*

*"Given the fact that deep convections sometimes develop from shallow convections (Houze 1993; Li and Schumacher, 2011; Yang et al., 2015), it is possible that the subjective compositing method will divide one precipitation event into different phases, which will lead to large uncertainties in determining precipitation regimes from TRMM data alone. This deserves more explicit analyses aided by geostationary satellite data in the future, which is out of the scope of this study."*

2. Page 16 line 4: Regarding the inference of aerosol radiative effect in "which may be partly due to the aerosol radiative", key reference is missing, for example, Liu et al. GRL 2018.

Liu Z., Yim S.H.L., Wang C., Lau N.C. (2018). The impact of the aerosol direct radiative forcing on deep convection and air quality in the Pearl River Delta region. Geophysical Research Letters, 45(9), 4410-4418"

*Response: The reference has been cited as suggested.*

---

## Author Comment (AC3) · 16 Aug 2018

**Response to Reviewer #2' Comments**

**Reviewer: Anonymous**

**General comments:**
Taking the Pearl River Delta region of China as the study area, this paper reported the observational evidence of aerosol effect on precipitation using TRMM and $PM_{10}$ datasets. Different precipitation types or regimes in particular the vertical structure of precipitation are differentiated for more explicit explanation. The findings from the study are convincing, and the technical details are sufficient to support the results. The impacts from other meteorological factors are also discussed in order to single out the aerosol effects on precipitation. The study is at the frontier of this research field, and I highly recommend the paper is published in ACP at current format. I only have a few minor comments and suggestions for the authors to consider if they are going to revise the paper further for improvements.
*Response: We thank the reviewer for his/her positive comments on our work. We have tried as much as possible to address all concerns and have revised the manuscript accordingly. The comments are written in normal font, and our point-to-point responses to the comments are in bold italics.*

**Minor comments:**
1. Page 6 L24: problems -> considerations?
*Response: Revised as suggested.*

2. Page 7 Line 25: tercile -> terciles?
*Response: Revised as suggested.*

3. P7L26: dirtiest -> most polluted?
*Response: Revised as suggested.*

4. P7L27: bins -> terciles?
*Response: Revised as suggested.*

5. Table 2: would it be possible to add $PM_{10}$ information in this table as well? Such as Mean and Standard Deviation of $PM_{10}$ for Clean and Polluted terciles?
*Response: Done as suggested.*

6. Page 8 L5-7: will such definitions of clean and polluted conditions match the terciles with each tercile having same number of samples?
*Response: Yes, the number of samples under the clean and polluted conditions are the same. In other words, we got the same numbers of localized precipitation events, instead of profiles, for further analysis, as detailed in Table 2. In order to clarify*

*this issue, more related discussion has been added here in this revised manuscript.*

7. Figure 1 caption: is $PM_{10}$ in (a) collected over the same period of $PM_{2.5}$? If not, it is better to be specific.

*Response: Yes, both of $PM_{10}$ and $PM_{2.5}$ measurements are taken during the same period from November 2013 to October 2014. We clarify this issue by revising Figure 1 caption: "Spatial distributions of (a) ground-based mean PM10 (in μg/m3) and (b) the ratio of mean PM2.5 to mean PM10 simultaneously measured for the period from November 2013 to October 2014. The red box outlines the PRD region, the dots show the locations of the PM measurement sites."*

8. Figure 2 reminds me how the seasonal cycles in the datasets are treated. For example, $PM_{10}$ are collected from different months, did you deseasonalize the datasets before the three terciles are determined? This will help avoid high tercile $PM_{10}$ are mostly collected from one season, and low tercile are from another season. It is therefore worthwhile and critical to check whether your samples from each tercile are biased to seasons.

*Response: We appreciate your critical and constructive suggestions. You are totally right, there exists some kind of seasonal bias in our samples. Therefore, we double check the seasonal distribution of the number of localized precipitation events. More importantly, the potential influence caused this possible bias has been discussed in this revised manuscript, by adding the following two new sub-sections in the Supporting Information, which is shown as follows:*

*"1. Seasonal distribution of the number of localized precipitating events*

*Figure S1 show the seasonal distribution with regard to the number of localized precipitating events used in the main text. For each rain regime, blue and yellow bars represent clean and polluted conditions, respectively. For the shallow regime, the numbers of localized events under clean/polluted conditions are 11/7, 9/6, 9/16, and 7/7 for spring, summer, autumn, and winter, respectively. By comparison, for stratiform regime, they are 7/3, 3/5, 5/5, and 4/6; for convective regime, they are 9/5, 6/8, 7/13, and 8/4, respectively. This indicates that there are not dramatic seasonal difference in terms of sampling. This is because our study region is in the tropics with much weaker seasonal variation than high latitudes.*

*4. Uncertainties of aerosol effect on the vertical structure of precipitation caused by sampling variation with seasons*

*To check its potential impact induced by the number of precipitation events by seasons (Figure S1), Figure S5 presents the analysis results in the same way as in Figure 5 in the main text, but for summer (June, July, and August). Note that the dominant precipitation falls over the PRD region in summer. The similar*

*differences of normalized contoured frequency by altitude diagram (ΔNCFAD) pattern as shown in Figure 5 suggest limited seasonal contamination. That is mainly because we focus on localized precipitation only, therefore strictly control on weather patterns are performed and large seasonal differences have been excluded. Furthermore, the potential influences of different dynamic and thermodynamic conditions during different seasons have been mostly revealed in section 3.4 in the main text. "*

*In addition, related description has also been added to discuss the potential effect induced by variation of samples with seasons in section 2.3.1.*

[Figure]

*Figure S1. Seasonal distribution of the number of localized precipitating events for spring (March, April, and May), summer (June, July, and August), autumn (September, October, and November), and winter (December, January, and February). For each precipitation regime, blue and yellow bars represent clean and polluted conditions.*

9. P13 L12: Figure4c -> Figure 4(c).
*Response: It has been revised to Figure 4c to make it consistent with other instances throughout our manuscript.*

10. $PM_{10}$ datasets availability: Did $PM_{10}$ data end in December 2012 and then replaced by $PM_{2.5}$ in 2013? Why we don't have any $PM_{10}$ any more after 2012 to extend the study period longer? It is strange they stopped measuring $PM_{10}$ after they switched to $PM_{2.5}$ in 2013.
*Response: Good point! Actually, there are simultaneous $PM_{2.5}$ and $PM_{10}$ observations after 2012. The study period stopped in the end of 2012 simply due to extremely large data volume acquired from Precipitation Radar onboard TRMM. More importantly, 6 years of data already show us robust results.*

11. Figure 6: did you conduct the similar thing for shallow and stratiform precipitation?
*Response: No, we did not do it, considering their relatively weak rain intensity. As shown in Figure 5a-b, either shallow or stratiform precipitation has relatively lower*

*radar reflectivity (less than 40 dBZ), compared with convective precipitation. In addition, it is hard for both of these two precipitation regimes to develop deeply to high altitudes. This further makes us not be able to do a similar analysis to convective precipitation.*

12. Figure 7 and 8: I am not sure I understand them completely. But I do like the discussions associated with them.

*Response: To improve the scientific importance and implications of Figures 7-8, we reorganized the structure of "Section 3. Results and discussion", please see our revised manuscript for more details.*

---

## Author Response (AR1)

**Response to Reviewers #1' Comments**

Reviewer: Y. Lin

**General comments:**
The study utilized the TRMM radar reflectivity and PM10 data over the PRD region to investigate the potential impacts of aerosol on precipitation. How to quantify aerosol impacts on precipitation based solely on observations is a tough task since meteorological factors need to be isolated effectively. The study is unique in that it separated precipitation associated with synoptic or mesoscale forcing from those localized precipitation events. Furthermore, other meteorological factors, including vertical wind shear, which is important for convective system development, are also analyzed and described. The finding that aerosol is able to invigorate deep convections is generally consistent with previous modeling studies. The study is thus a good contribution to this community. Nevertheless, I have some suggestions for the authors to consider.
*Response: We thank the reviewer for his thoughtful and thorough comments and suggestions. We have tried as much as possible to address all concerns and have revised the manuscript accordingly. The comments are written in normal font, and our point-to-point responses to the comments are in bold italics.*

**Specific comments:**
1. Although manual identification of synoptic or localized precipitation event is described on 26-28 p11, a few more description might help since it is very subjective. I am also wondering whether localized precipitation is more appropriate than local-scale precipitation.
*Response: Thanks for pointing this out. More descriptions regarding how to identify a synoptic or localized precipitation event are added in Section 3.1 of our revised manuscript, which are described as follows:*

*"…The discrimination between localized and synoptic-scale precipitation events for a given day largely relies on the weather composite charts, where daily averaged wind field at 850 hPa was overlaid with the geopotential height at 500hPa. Particularly, the localized precipitation event for a given day was subjectively determined as follows: (1) There exists favorable atmospheric conditions for the initiation and development of localized precipitation events through visual interpretation of the weather composite plot for the day analyzed; (2) The minimum rainfall greater than 0.1 mm/d was recorded at any gauges in the study area (red box in Figure 1); (3) there are ground-based $PM_{10}$ measurements collocated with precipitation measurements from TRMM to obtain a pair of valid aerosol-precipitation data. As such, the total number of collocated samples reached up to 253 for localized precipitation events, whereas 194 for synoptic scale*

*precipitation events…".*

*In addition, "local-scale precipitation" has been revised to "localized precipitation", per your suggestion.*

2. Smaller reflectivity below the freezing level for polluted cases than clean cases (Fig. 5c, P14 Line 10) might be due to the large numbers, but smaller sizes of rain drops within polluted environment.
*Response: Agreed, and we add the following discussion to better elucidate the possible causes for the smaller reflectivity observed below the freezing level under polluted conditions:*
*"Below the freezing level where the reflectivity is less than 40 dBZ, the color is virtually all blue, meaning that precipitation is weaker under polluted conditions than clean ones. This could also be due to a large number of smaller rain drops within polluted environment."*

3. Looks like $PM_{10}$ (P8, Line 5) is much higher during the periods with occurrence of shallow convection than other two types of precipitation. Any reasons for this? Does this imply heavy pollution tends to inhibit deep convection development sometimes, although it will invigorate deep convection once the negative impacts of aerosols are overcome?
*Response: Agreed. The phenomenon you noticed likely imply heavy pollution tends to inhibit deep convection development sometimes. After double checking the original dataset, one cause for such a higher average PM10 for shallow precipitation regime is due to two days with abnormal high $PM_{10}$ concentration, corresponding to 255.33 and $260\mu g/m^3$, respectively. In contrast, for other two precipitation regimes (i.e., Stratiform and convective), the maximal $PM_{10}$ concentration is just $193\mu g/m^3$. As you suggested, this implies heavy pollution tends to inhibit deep convection development sometimes, although it will invigorate deep convection once the negative impacts of aerosols are overcome. Related discussion has been added to our revised manuscript.*

4. Regarding that deep convections sometimes developed from shallow convections, is it possible that the composite will divide one precipitation event into different types. This need to be mentioned somehow.
*Response: Per your suggestions, we added the following discussion in section 3.1:*

*"Given the fact that deep convections sometimes develop from shallow convections (Houze 1993; Li and Schumacher, 2011; Yang et al., 2015), it is possible that the subjective composite method will divide one precipitation event into different types, which will lead to large uncertainties in determining precipitation regimes from TRMM data alone. This deserves more explicit analyses aided by geostationary satellite data in the future, which is out of the scope of this study."*

**Minor comments:**

1. Why use the vertical wind shear between 1000 and 700 hPa instead over a higher level?

*Response: This is a typo, since we confused the two pressure levels used to define the wind shear with those for the calculation of LTS (lower troposphere stability). LTS is defined as potential temperature difference between 1000hPa and 700hPa. Actually, the vertical wind shear used in the main text is calculated from the winds between 850hPa (~1.5km) and 500hPa (~5.5 km), rather than between 1000hPa and 700hPa, which has been corrected in this revised manuscript.*

2. P6, L20, delete "use to"

*Response: Deleted as suggested.*

3. There are some other typos. Please double check.

*Response: We corrected other typos in our revised manuscript.*

[revised manuscript text omitted]

Formatted ... [27]
Formatted ... [26]
Formatted ... [28]
Formatted ... [30]
Formatted ... [32]
Formatted ... [33]
Moved up [1]: Nakajima et al., 2010;Suzuki et al.,
Formatted ... [34]
Formatted ... [35]
Moved (insertion) [1]
Formatted ... [37]

Measuring Mission (TRMM) and ground-based in-situ aerosol measurements made in the Pearl River Delta (PRD) region of southern China. We will examine differences in the vertical structure of precipitation between clean and polluted atmospheric environments to determine whether they are consistent with some previously proposed mechanisms governing aerosol invigoration or suppression of precipitation.

The rest of this paper proceeds as follows. The study area, datasets, and methods used here are described in section 2. How to discriminate between synapitical-scale and locallized precipitating systems, the potential aerosol-induced changes in the vertical structure of different precipitation regimes and their dependences on meteorological conditions are discussed in section 3. Finally, the main findings of this study are summarized in section 4.

**2 Data and methods**

**2.1 Study area**

The study area is mainly over the PRD region (bounded by 113°E and 115°E, 22°N and 24°N, red rectangles in Figure 1), including many populated cities with relatively high emissions (e.g., Guangzhou, Shenzhen, Zhuhai, and Hong Kong). The PRD has a humid subtropical climate, which is strongly influenced by the Asian monsoon circulation and tropical cyclones originated in the western Pacific Ocean (Ding, 1994). In recent decades, the PRD region experienced rapid economic development, which caused heavy air pollutions associated with human activities, including the increasing fossil fuel combustion due to industrialization (Deng et al., 2008; Guo et al., 2009; Guo et al., 2016b). In addition, another main reason for us to take the PRD region as our region of interest (ROI) is the well documented significant positive correlations between air pollution and occurrence frequency of precipitation over this area (e.g., Wang et al., 2011; Yang and Li, 2014).

**2.2 Data**

The dataset used here are listed in Table 1 and are briefly described here. Notably, six years (from 1 January, 2007 to 31 December, 2012, unless noted otherwise) of precipitation measurements from the TRMM PR (version 7, Huffman et al., 2007), combined with collocated aerosol data collected at ground surface, and meteorological data from the European Centre for

Formatted … [46]
Formatted … [47]
Formatted … [55]
Formatted … [59]
Formatted … [60]
Formatted … [64]
Moved (insertion) [2]

Medium-Range Weather Forecasts (ECMWF) ERA-Interim reanalysis (Dee et al., 2011) are analyzed here. Prior to further explicit observational analyses, the spurious signals likely resulting from measurement uncertainties should be firstly considered, such as the misclassification of rain profiles, abnormal observations, and so on. To minimize such uncertainties, we screen the aerosol and precipitation observational data very carefully, which will be detailed as follows.

*2.2.1 TRMM PR and 3B42 data*

The precipitation properties are obtained from the TRMM PR products 2A25 and 3B42 (Huffman et al., 2007). For each rain profile, the information of category, attenuation-corrected reflectivity (Z) and rain rate (R) are provided by 2A25 with a vertical/horizontal resolution ~250 m/~4-5 km, depending on the satellite orbit height and the PR off-nadir view angle. The profile ranges from the near-surface to 20km altitude. 2A25 classify each rain profile as convective or stratiform rain with different confidence levels. Here we obtain rain profiles identified as stratiform or convective precipitation based on the 2A25 products alone, and further extract the shallow isolated echo category from convective precipitation as shallow regimes for better characterizing precipitating system. The classification is done for each profile, so different rain regimes could come from the same precipitation event. Their possible dynamic and thermodynamic connections, therefore, likely cause certain uncertainties in the following analyses, which will be discussed later. Additionally, two criteria are used to ensure that each profile contains a reliable precipitation event: (1) $Z \geq 15$ dBZ (the minimum detectable Z for the TRMM PR, Kummerow et al., 1998); and (2) at least four consecutive levels with $Z \geq 15$ dBZ are required for each profile. The horizontal distribution of R is provided by 3B42 with a spatial/temporal resolution of 0.25° x 0.25°/3-hourly over the global belt between 50°N and 50°S. 3B42 merges precipitation radar and microwave rainfall estimates with infrared-based precipitation estimates from multiple satellites, as well as measurements from rain gauges (Huffman et al., 2007).

*2.2.2 Ground-based PM₁₀ measurements*

Given the difficulties in obtaining large-scale CCN concentration information, we have to resort to any CCN proxy such as satellite-derived AOD and the aerosol index (AI), or ground-based particulate matter (PM) measurements. Sound correlations have been extensively documented between satellite retrievals of AOD, and cloud and precipitation properties (e.g., Koren et al., 2005, 2012; Huang et al., 2009b). Such correlations, however, are susceptible to

Formatted … [65]
Formatted … [66]
Formatted … [68]
Formatted … [70]
Moved up [2]: the European Centre for Medium-Range
Formatted … [72]
Formatted … [76]
Formatted … [73]
Formatted … [84]
Formatted … [85]
Formatted … [91]

[revised manuscript text omitted]

Font: Not Bold, Not Italic, (Asian) Chinese (China)

| Page 2: [2] Deleted | huan | 8/5/18 2:01:00 PM |
|---|---|---|

higher

| Page 2: [3] Deleted | Jianping Guo | 8/8/18 7:41:00 AM |
|---|---|---|

.

| Page 2: [3] Deleted | Jianping Guo | 8/8/18 7:41:00 AM |
|---|---|---|

.

| Page 2: [3] Deleted | Jianping Guo | 8/8/18 7:41:00 AM |
|---|---|---|

.

| Page 2: [3] Deleted | Jianping Guo | 8/8/18 7:41:00 AM |
|---|---|---|

.

| Page 2: [3] Deleted | Jianping Guo | 8/8/18 7:41:00 AM |
|---|---|---|

.

| Page 2: [3] Deleted | Jianping Guo | 8/8/18 7:41:00 AM |
|---|---|---|

.

| Page 2: [3] Deleted | Jianping Guo | 8/8/18 7:41:00 AM |
|---|---|---|

.

| Page 2: [3] Deleted | Jianping Guo | 8/8/18 7:41:00 AM |
|---|---|---|

.

| Page 2: [3] Deleted | Jianping Guo | 8/8/18 7:41:00 AM |
|---|---|---|

.

| Page 2: [4] Deleted | huan | 7/16/18 1:54:00 PM |
|---|---|---|

. Radar reflectivity of the top 1% increases as the atmosphere becomes slightly polluted ($PM_{10}<38$ μg/m$^3$), except for shallow convection. The aerosol-precipitation data pairs are further limited to local- or meso-scalelocalized[HL1] precipitation systems. Results show that significant changes in precipitation

| Page 2: [5] Deleted | Jianping Guo | 8/8/18 7:23:00 AM |

In

| Page 2: [5] Deleted | Jianping Guo | 8/8/18 7:23:00 AM |

In

| Page 2: [5] Deleted | Jianping Guo | 8/8/18 7:23:00 AM |

In

| Page 2: [5] Deleted | Jianping Guo | 8/8/18 7:23:00 AM |

In

| Page 2: [5] Deleted | Jianping Guo | 8/8/18 7:23:00 AM |

In

| Page 2: [5] Deleted | Jianping Guo | 8/8/18 7:23:00 AM |

In

| Page 2: [5] Deleted | Jianping Guo | 8/8/18 7:23:00 AM |

In

| Page 2: [5] Deleted | Jianping Guo | 8/8/18 7:23:00 AM |

In

| Page 2: [5] Deleted | Jianping Guo | 8/8/18 7:23:00 AM |

In

| Page 2: [5] Deleted | Jianping Guo | 8/8/18 7:23:00 AM |

In

| Page 2: [5] Deleted | Jianping Guo | 8/8/18 7:23:00 AM |

In

| Page 2: [6] Deleted | huan | 7/16/18 2:01:00 PM |

are possibly induced by aerosol, and this potential aerosol effect is regime dependent. The

| Page 2: [6] Deleted | huan | 7/16/18 2:01:00 PM |

are possibly induced by aerosol, and this potential aerosol effect is regime dependent. The

| Page 2: [6] Deleted | huan | 7/16/18 2:01:00 PM |

are possibly induced by aerosol, and this potential aerosol effect is regime dependent. The

| Page 2: [6] Deleted | huan | 7/16/18 2:01:00 PM |

are possibly induced by aerosol, and this potential aerosol effect is regime dependent. The

| Page 2: [6] Deleted | huan | 7/16/18 2:01:00 PM |
|---|---|---|

are possibly induced by aerosol, and this potential aerosol effect is regime dependent. The

| Page 2: [7] Deleted | Liu, Huan | 7/2/18 5:37:00 PM |
|---|---|---|

18.7

| Page 2: [7] Deleted | Liu, Huan | 7/2/18 5:37:00 PM |
|---|---|---|

18.7

| Page 2: [7] Deleted | Liu, Huan | 7/2/18 5:37:00 PM |
|---|---|---|

18.7

| Page 2: [8] Deleted | huan | 7/16/18 2:26:00 PM |
|---|---|---|

precipitation

| Page 2: [8] Deleted | huan | 7/16/18 2:26:00 PM |
|---|---|---|

precipitation

| Page 2: [9] Deleted | Jianping Guo | 8/8/18 7:26:00 AM |
|---|---|---|

,

| Page 2: [9] Deleted | Jianping Guo | 8/8/18 7:26:00 AM |
|---|---|---|

,

| Page 2: [9] Deleted | Jianping Guo | 8/8/18 7:26:00 AM |
|---|---|---|

,

| Page 2: [10] Deleted | Jianping Guo | 8/11/18 11:01:00 PM |
|---|---|---|

rain

| Page 2: [10] Deleted | Jianping Guo | 8/11/18 11:01:00 PM |
|---|---|---|

rain

| Page 2: [11] Deleted | huan | 7/16/18 2:32:00 PM |
|---|---|---|

smaller (~10%[HL2])and inconformitybetween pristine and polluted conditions

| Page 2: [11] Deleted | huan | 7/16/18 2:32:00 PM |
|---|---|---|

smaller (~10%[HL3])and inconformitybetween pristine and polluted conditions

| Page 2: [11] Deleted | huan | 7/16/18 2:32:00 PM |
|---|---|---|

smaller (~10%[HL4])and inconformitybetween pristine and polluted conditions

| Page 2: [11] Deleted | huan | 7/16/18 2:32:00 PM |
|---|---|---|

smaller (~10%[HL5])and inconformitybetween pristine and polluted conditions

| Page 2: [11] Deleted | huan | 7/16/18 2:32:00 PM |
|---|---|---|

smaller (~10%[HL6])and inconformitybetween pristine and polluted conditions

| Page 2: [12] Deleted | Jianping Guo | 8/8/18 7:36:00 AM |
|---|---|---|

Dynamic and thermodynamic conditions which favor intensity precipitation always link with much positive aerosol effects

| Page 2: [13] Deleted | huan | 8/5/18 2:21:00 PM |
|---|---|---|

hypothesis

| Page 3: [14] Deleted | Jianping Guo | 8/11/18 10:05:00 PM |
|---|---|---|

Interactions between c

| Page 3: [14] Deleted | Jianping Guo | 8/11/18 10:05:00 PM |

Interactions between c

| Page 3: [15] Deleted | huan | 7/16/18 6:20:00 PM |

have been

| Page 3: [16] Deleted | Jianping Guo | 8/8/18 8:15:00 AM |

showing

| Page 3: [17] Deleted | huan | 7/16/18 2:48:00 PM |

l

| Page 3: [18] Deleted | Jianping Guo | 8/8/18 8:16:00 AM |

Tao et al., 2012;

| Page 3: [18] Deleted | Jianping Guo | 8/8/18 8:16:00 AM |

Tao et al., 2012;

| Page 3: [19] Deleted | Jianping Guo | 8/8/18 8:20:00 AM |

therefore

| Page 3: [19] Deleted | Jianping Guo | 8/8/18 8:20:00 AM |

therefore

| Page 3: [19] Deleted | Jianping Guo | 8/8/18 8:20:00 AM |

therefore

| Page 3: [20] Formatted | Jianping Guo | 8/11/18 11:29:00 PM |

Not Highlight

| Page 3: [20] Formatted | Jianping Guo | 8/11/18 11:29:00 PM |

Not Highlight

| Page 3: [21] Deleted | Jianping Guo | 8/8/18 8:20:00 AM |

, which

| Page 3: [21] Deleted | Jianping Guo | 8/8/18 8:20:00 AM |

, which

| Page 3: [22] Deleted | Jianping Guo | 8/8/18 8:21:00 AM |

;

| Page 3: [22] Deleted | Jianping Guo | 8/8/18 8:21:00 AM |

;

| Page 3: [22] Deleted | Jianping Guo | 8/8/18 8:21:00 AM |

;

| Page 3: [22] Deleted | Jianping Guo | 8/8/18 8:21:00 AM |

;

| Page 3: [23] Deleted | huan | 7/16/18 2:53:00 PM |

Convective iAerosol invigoration effects on deep convections has been suggestedin ample studies that includingboth the cloud top height

| Page 3: [23] Deleted | huan | 7/16/18 2:53:00 PM |

| Page 3: [23] Deleted | huan | 7/16/18 2:53:00 PM |

| Page 3: [23] Deleted | huan | 7/16/18 2:53:00 PM |

| Page 3: [23] Deleted | huan | 7/16/18 2:53:00 PM |

| Page 3: [23] Deleted | huan | 7/16/18 2:53:00 PM |

| Page 3: [24] Deleted | Jianping Guo | 8/8/18 8:22:00 AM |

, which

| Page 3: [24] Deleted | Jianping Guo | 8/8/18 8:22:00 AM |

, which

| Page 4: [25] Deleted | Jianping Guo | 8/8/18 8:37:00 AM |

Aerosol microphysical effects can fuel competitive microphysical processes at the same time, therefore play different even opposite roles on clouds and precipitation under different meteorological conditions (Khain et al., 2008; Fan et al., 2009; Li et al., 2011; Dagan et al., 2015). Furthermore, smaller droplets of higher motilities, slower freezing processes, and stronger entrainment (Koren et al., 2015; Rosenfeld and Woodley, 2000; Pinsky et al., 2013;), which can link the aerosol effects on microphysical processes to thermodynamic conditions, therefore further fuel or consume clouds' energy budget, and influence precipitation (Koren et al., 2005; Rosenfeld et al., 2008; Koren et al., 2012).

| Page 4: [26] Formatted | Jianping Guo | 8/11/18 11:29:00 PM |

Not Highlight

| Page 4: [27] Formatted | Jianping Guo | 8/11/18 11:29:00 PM |

Not Highlight

| Page 4: [28] Formatted | Jianping Guo | 8/11/18 11:29:00 PM |

Not Highlight

| Page 4: [29] Deleted | huan | 7/16/18 6:56:00 PM |

At the same time, the inhibition of light precipitationby aerosolshas also been reported indifferent regions of overthe world(Kaufman and Fraser, 1997; Rosenfeld and Lensky, 1998; Rosenfeld and Givati, 2006; Wang et al., 2011; Guo et al., 2014). The invigoration theory was recently generalized by Fan et al. (2018) that can also occur for shallower water clouds under extreme clean conditions under which ultra-fine mode aerosol particles may be nucleated to release latent heat to fuel cloud development. While we have come a long way in understanding the mechanisms behind various observation-based findings, the impact of aerosol on precipitation remain a daunting task(Tao et al, 2012). Failure in fully understanding and accounting for these effects may not only undermine our understanding of the earth's climate and its changes (IPCC, 2013), but also impairthe accuracy of rainfall forecast by a numerical weather model (Jiang et al., 2017).

The specific effects of aerosols on precipitation are strongly influenced and confounded by atmospheric dynamic and thermodynamic conditions, such as updraft strength (Koren et al., 2012;

Tao et al., 2012; Guo et al., 2016), wind shear (Fan et al., 2009), and atmospheric instability (Gordon, 1994;Khain et al., 2004). By serving as cloud condensation nuclei (CCN), high aerosol concentrationleads to more butsmaller cloud droplets that consume the available water vapor more efficiently(Koren et al., 2010). Consequently, aerosols can indirectly modify the vertical profiles of hydrometeors and cloud phases, which can,in turn, alter the dynamics and thermodynamics of a precipitating cloud system through latent heat releasinge(Heiblum etal., 2012). Relationships between aerosols and precipitation also vary significantlyon seasonal and spatial scales(Huang et al., 2009a,b,c). It has been a great challenge tosingle out the aerosol effects,largely due to various processes influencing precipitation, radiation, and even the state of the atmosphere that areinduced by aerosols.

| Page 4: [30] Formatted | Jianping Guo | 8/11/18 11:29:00 PM |
|---|---|---|

Font: (Asian) +Body Asian (SimSun)

| Page 4: [31] Deleted | huan | 7/16/18 7:24:00 PM |
|---|---|---|

on, which is determined by a combination of dynamic, thermodynamic, and microphysical processesoccurring in precipitation systems

| Page 4: [32] Formatted | Jianping Guo | 8/11/18 11:29:00 PM |
|---|---|---|

Not Highlight

| Page 4: [33] Formatted | Jianping Guo | 8/11/18 11:29:00 PM |
|---|---|---|

Not Highlight

| Page 4: [34] Formatted | Jianping Guo | 8/11/18 11:29:00 PM |
|---|---|---|

Not Highlight

| Page 4: [35] Formatted | Jianping Guo | 8/11/18 11:29:00 PM |
|---|---|---|

Not Highlight

| Page 4: [36] Deleted | huan | 7/24/18 4:48:00 PM |
|---|---|---|

Nakajima et al., 2010; Suzuki et al., 2010; Chen et al., 2016

| Page 4: [37] Formatted | Jianping Guo | 8/11/18 11:29:00 PM |
|---|---|---|

Not Highlight

| Page 4: [38] Deleted | huan | 7/16/18 7:26:00 PM |
|---|---|---|

insights into themechanism underlying the aerosol-cloud-precipitation interaction mechanism. The deployment of the cloud profiling radar onboardCloudSat has indeed led to new insights into

cloud andprecipitation microphysical processes (e.g.,Nakajima et al., 2010;Suzuki et al., 2010; Chen et al., 2016). Studies examining aerosol effects on precipitation systemsusing satellite observations (e.g.,Rosenfeld, 2000; Niu and Li, 2012; Peng et al., 2016) are often limited to column-integratedaerosol optical depth (AOD) and cloud topproperties.

Given the dominant effects of cloud dynamics on synoptic-scale precipitation systems, only precipitation events occurring on local scale are examined in detail in the following sections. This consideration is largely due to local- or meso-scalelocalized precipitating clouds, including the thermal convection, cumulus, and stratocumulus clouds, are less dependent on large scale dynamic conditions and more susceptible to aerosol pollution (Fan et al., 2007; Lee et al., 2012; Guo et al., 2017).
* * *
**Page 4: [39] Deleted**      **Jianping Guo**      **8/11/18 10:40:00 PM**

Studies examining aerosol effects on precipitation systems using satellite observations (e.g., Rosenfeld, 2000; Niu and Li, 2012; Peng et al., 2016) are often limited to column-integrated aerosol optical depth (AOD) and cloud top properties.
* * *
**Page 4: [40] Deleted**      **Jianping Guo**      **8/11/18 10:46:00 PM**

s, including the thermal convection, cumulus, and stratocumulus clouds, are less dependent on large scale dynamic conditions and more susceptible to
* * *
**Page 5: [41] Deleted**      **Jianping Guo**      **8/11/18 10:47:00 PM**

We will examine differences in the vertical structureof precipitationbetween relatively clean and dirty atmospheric environments to determine whether they areconsistent withsome previously proposed mechanisms governingaerosol invigoration or suppression of precipitation.Note that, by considering the representativeness of in-situ aerosol measurements and higher sensitivity of rain systems to local atmospheric properties (Guo et al., 2017; Lin et al., 2018), only localized precipitating events (such as thermal convections) are deeply analyzed.
* * *
**Page 5: [42] Deleted**      **Jianping Guo**      **8/8/18 8:45:00 AM**

S
* * *
**Page 5: [42] Deleted**      **Jianping Guo**      **8/8/18 8:45:00 AM**

S

| Page 5: [45] Deleted | huan | 7/28/18 9:48:00 AM |
|---|---|---|

we examine any dependenceof the vertical structure of precipitationon aerosols

| Page 5: [46] Formatted | Jianping Guo | 8/11/18 11:29:00 PM |
|---|---|---|

Font: Bold

| Page 5: [47] Formatted | Jianping Guo | 8/11/18 11:29:00 PM |
|---|---|---|

Font: Bold

| Page 5: [48] Deleted | huan | 8/5/18 3:42:00 PM |
|---|---|---|

region of interest is the

| Page 5: [48] Deleted | huan | 8/5/18 3:42:00 PM |
|---|---|---|

region of interest is the

| Page 5: [49] Deleted | Liu, Huan | 7/3/18 8:34:00 AM |
|---|---|---|

-

| Page 5: [49] Deleted | Liu, Huan | 7/3/18 8:34:00 AM |
|---|---|---|

-

| Page 5: [49] Deleted | Liu, Huan | 7/3/18 8:34:00 AM |
|---|---|---|

-

| Page 5: [50] Deleted | huan | 8/6/18 4:27:00 PM |
|---|---|---|

ure

| Page 5: [50] Deleted | huan | 8/6/18 4:27:00 PM |
|---|---|---|

ure

| Page 5: [51] Deleted | huan | 8/6/18 4:30:00 PM |
|---|---|---|

such as

| Page 5: [51] Deleted | huan | 8/6/18 4:30:00 PM |
|---|---|---|

such as

| Page 5: [52] Deleted | huan | 8/5/18 3:43:00 PM |
|---|---|---|

,

| Page 5: [52] Deleted | huan | 8/5/18 3:43:00 PM |
|---|---|---|

,

| Page 5: [52] Deleted | huan | 8/5/18 3:43:00 PM |
|---|---|---|

,

| Page 5: [53] Deleted | huan | 8/5/18 3:44:00 PM |
|---|---|---|

in recent years.

| Page 5: [53] Deleted | huan | 8/5/18 3:44:00 PM |
|---|---|---|

in recent years.

| Page 5: [53] Deleted | huan | 8/5/18 3:44:00 PM |
|---|---|---|

in recent years.

| Page 5: [54] Deleted | Jianping Guo | 8/11/18 10:55:00 PM |
|---|---|---|

(e.g.,

| Page 5: [54] Deleted | Jianping Guo | 8/11/18 10:55:00 PM |
|---|---|---|

(e.g.,

| Page 5: [55] Formatted | Jianping Guo | 8/11/18 11:29:00 PM |
|---|---|---|

Not Highlight

| Page 5: [56] Deleted | Jianping Guo | 8/9/18 7:00:00 AM |
|---|---|---|

Except the characters

| Page 5: [56] Deleted | Jianping Guo | 8/9/18 7:00:00 AM |
|---|---|---|

Except the characters

| Page 5: [57] Deleted | huan | 7/17/18 11:12:00 AM |
|---|---|---|

Observations have shown positive correlations between air pollution levels and that precipitation and the frequency of lightningfrequencieshave been enhanced in recent years in southern China

| Page 5: [58] Deleted | Liu, Huan | 7/3/18 8:39:00 AM |
|---|---|---|

, as atmospheric pollution worsened in the region

| Page 5: [59] Formatted | Jianping Guo | 8/11/18 11:29:00 PM |
|---|---|---|

Not Highlight

| Page 5: [59] Formatted | Jianping Guo | 8/11/18 11:29:00 PM |
|---|---|---|

Not Highlight

| Page 5: [60] Formatted | Jianping Guo | 8/11/18 11:29:00 PM |
|---|---|---|

Font: Bold

| Page 5: [61] Deleted | huan | 8/6/18 4:31:00 PM |
|---|---|---|

s

| Page 5: [61] Deleted | huan | 8/6/18 4:31:00 PM |
|---|---|---|

s

| Page 5: [62] Deleted | Jianping Guo | 8/9/18 7:14:00 AM |
|---|---|---|

.

| Page 5: [62] Deleted | Jianping Guo | 8/9/18 7:14:00 AM |
|---|---|---|

.

| Page 5: [62] Deleted | Jianping Guo | 8/9/18 7:14:00 AM |
|---|---|---|

.

| Page 5: [63] Deleted | huan | 7/17/18 11:19:00 AM |
|---|---|---|

 and

| Page 5: [63] Deleted | huan | 7/17/18 11:19:00 AM |
|---|---|---|

 and

| Page 5: [64] Formatted | Jianping Guo | 8/11/18 11:29:00 PM |
|---|---|---|

Not Highlight

| Page 6: [65] Formatted | Jianping Guo | 8/11/18 11:29:00 PM |
|---|---|---|

Not Highlight

| Page 6: [65] Formatted | Jianping Guo | 8/11/18 11:29:00 PM |
|---|---|---|

Not Highlight

| Page 6: [66] Formatted | Jianping Guo | 8/11/18 11:29:00 PM |
|---|---|---|

Font color: Red

| Page 6: [67] Deleted | huan | 7/17/18 11:19:00 AM |
|---|---|---|

.

| Page 6: [67] Deleted | huan | 7/17/18 11:19:00 AM |
|---|---|---|

.

| Page 6: [68] Formatted | Jianping Guo | 8/11/18 11:29:00 PM |
|---|---|---|

Font: Times New Roman

| Page 6: [69] Deleted | Jianping Guo | 8/3/18 8:58:00 AM |
|---|---|---|

To

| Page 6: [69] Deleted | Jianping Guo | 8/3/18 8:58:00 AM |
|---|---|---|

To

| Page 6: [70] Formatted | Jianping Guo | 8/11/18 11:29:00 PM |
|---|---|---|

Font: (Default) Times New Roman

| Page 6: [71] Deleted | Jianping Guo | 8/3/18 8:59:00 AM |
|---|---|---|

data retrieval

| Page 6: [71] Deleted | Jianping Guo | 8/3/18 8:59:00 AM |
|---|---|---|

data retrieval

| Page 6: [71] Deleted | Jianping Guo | 8/3/18 8:59:00 AM |
|---|---|---|

data retrieval

| Page 6: [71] Deleted | Jianping Guo | 8/3/18 8:59:00 AM |
|---|---|---|

data retrieval

| Page 6: [72] Formatted | Jianping Guo | 8/11/18 11:29:00 PM |
|---|---|---|

Font: (Default) Times New Roman

| Page 6: [73] Formatted | Jianping Guo | 8/11/18 11:29:00 PM |
|---|---|---|

Font: (Default) Times New Roman

| Page 6: [74] Deleted | Liu, Huan | 7/3/18 9:03:00 AM |
|---|---|---|

Aerosol loading informationretrieved by space-borne sensors islimitedto cloud-free conditions, leading to a lack ofcoincidentand collocated measurements of aerosols and precipitation.

| Page 6: [75] Deleted | huan | 7/17/18 11:23:00 AM |
|---|---|---|

The pParticulate matter (PM) with an aerodynamic diameter less than 10 μm (PM$_{10}$, limited to ≤ 200 μ g/m$^3$) measured at surface is thusused as a proxy of aerosol loading[HL7], **μ g/m$^3$** to exclude abnormally samples. Meteorological variables are taken from

| Page 6: [76] Formatted | Jianping Guo | 8/11/18 11:29:00 PM |
|---|---|---|

Superscript

| Page 6: [77] Deleted | huan | 7/17/18 11:23:00 AM |
|---|---|---|

To observational analyses, the possible artificial from data retrieval should be firstly considered, such as the misclassification of rain profiles and abnormal observations. In order to minimize such uncertainties, we filter our dataset carefully and tried our best in the statistical way as described as follows.

| Page 6: [78] Deleted | huan | 7/17/18 11:26:00 AM |
|---|---|---|

3D structures

| Page 6: [78] Deleted | huan | 7/17/18 11:26:00 AM |
|---|---|---|

3D structures

| Page 6: [78] Deleted | huan | 7/17/18 11:26:00 AM |
|---|---|---|

3D structures

| Page 6: [78] Deleted | huan | 7/17/18 11:26:00 AM |
|---|---|---|

3D structures

| Page 6: [78] Deleted | huan | 7/17/18 11:26:00 AM |
|---|---|---|

3D structures

| Page 6: [78] Deleted | huan | 7/17/18 11:26:00 AM |
|---|---|---|

3D structures

| Page 6: [78] Deleted | huan | 7/17/18 11:26:00 AM |

3D structures

| Page 6: [79] Deleted | Jianping Guo | 8/9/18 10:19:00 AM |

(

| Page 6: [79] Deleted | Jianping Guo | 8/9/18 10:19:00 AM |

(

| Page 6: [80] Deleted | huan | 7/24/18 6:10:00 PM |

into

| Page 6: [80] Deleted | huan | 7/24/18 6:10:00 PM |

into

| Page 6: [81] Deleted | Jianping Guo | 8/9/18 10:20:00 AM |

certain

| Page 6: [81] Deleted | Jianping Guo | 8/9/18 10:20:00 AM |

certain

| Page 6: [81] Deleted | Jianping Guo | 8/9/18 10:20:00 AM |

certain

| Page 6: [81] Deleted | Jianping Guo | 8/9/18 10:20:00 AM |

certain

| Page 6: [82] Deleted | huan | 7/17/18 11:31:00 AM |

types asprovided in the 2A23 product (profiles defined as certain rain type only). In order to

| Page 6: [82] Deleted | huan | 7/17/18 11:31:00 AM |
|---|---|---|

types asprovided in the 2A23 product (profiles defined as certain rain type only). In order to

| Page 6: [83] Deleted | Jianping Guo | 8/9/18 10:21:00 AM |
|---|---|---|

Note that, because t

| Page 6: [84] Formatted | Jianping Guo | 8/11/18 11:29:00 PM |
|---|---|---|

Font: (Default) Times New Roman

| Page 6: [84] Formatted | Jianping Guo | 8/11/18 11:29:00 PM |
|---|---|---|

Font: (Default) Times New Roman

| Page 6: [85] Formatted | Jianping Guo | 8/11/18 11:29:00 PM |
|---|---|---|

Font: (Default) Times New Roman

| Page 6: [85] Formatted | Jianping Guo | 8/11/18 11:29:00 PM |
|---|---|---|

Font: (Default) Times New Roman

| Page 6: [85] Formatted | Jianping Guo | 8/11/18 11:29:00 PM |
|---|---|---|

Font: (Default) Times New Roman

| Page 6: [86] Deleted | Jianping Guo | 8/9/18 10:22:00 AM |
|---|---|---|

could

| Page 6: [86] Deleted | Jianping Guo | 8/9/18 10:22:00 AM |
|---|---|---|

could

| Page 6: [86] Deleted | Jianping Guo | 8/9/18 10:22:00 AM |
|---|---|---|

could

| Page 6: [86] Deleted | Jianping Guo | 8/9/18 10:22:00 AM |
|---|---|---|

could

| Page 6: [87] Deleted | huan | 7/24/18 6:14:00 PM |
|---|---|---|

(detailed in section 3.2) and its association with aerosols, a third precipitation type, namely shallowprecipitation type, is included in this study.

| Page 6: [88] Deleted | Liu, Huan | 7/3/18 8:49:00 AM |
|---|---|---|

All pixels that do not exceed theradar reflectivitythreshold of 15 dBZ (the minimum detectable reflectivity factor forthe TRMM PR) are omitted (Kummerow et al., 1998).

| Page 6: [89] Deleted | huan | 7/24/18 6:21:00 PM |
|---|---|---|

T

| Page 6: [90] Deleted | huan | 7/17/18 11:41:00 AM |
|---|---|---|

the attenuation-corrected reflectivity (

the attenuation-corrected reflectivity (

| Page 6: [90] Deleted | huan | 7/17/18 11:41:00 AM |
|---|---|---|

the attenuation-corrected reflectivity (

| Page 6: [90] Deleted | huan | 7/17/18 11:41:00 AM |
|---|---|---|

the attenuation-corrected reflectivity (

| Page 6: [90] Deleted | huan | 7/17/18 11:41:00 AM |
|---|---|---|

the attenuation-corrected reflectivity (

| Page 6: [90] Deleted | huan | 7/17/18 11:41:00 AM |
|---|---|---|

the attenuation-corrected reflectivity (

| Page 6: [91] Formatted | Jianping Guo | 8/11/18 11:29:00 PM |
|---|---|---|

Superscript

| Page 6: [92] Deleted | huan | 8/5/18 5:06:00 PM |
|---|---|---|

there must be

| Page 6: [92] Deleted | huan | 8/5/18 5:06:00 PM |
|---|---|---|

there must be

| Page 6: [92] Deleted | huan | 8/5/18 5:06:00 PM |
|---|---|---|

there must be

| Page 6: [92] Deleted | huan | 8/5/18 5:06:00 PM |
|---|---|---|

there must be

| Page 6: [93] Deleted | Jianping Guo | 8/9/18 10:24:00 AM |
|---|---|---|

| Page 6: [93] Deleted | Jianping Guo | 8/9/18 10:24:00 AM |
|---|---|---|

| Page 6: [94] Deleted | huan | 7/24/18 6:26:00 PM |

The estimates are gridded at a 0.25°x0.25° spatial resolution over the global belt between 50°N and 50°S and have a three-hour temporal resolution.

| Page 7: [95] Deleted | Jianping Guo | 8/9/18 9:50:00 PM |

Sound correlations between AOD and CCN Previous studiesare reported (e.g., Koren et al., 2005, 2012; Jiang et al., 2008Andreae, 2009; Huang et al., 2009b).haveshown that thereare sound correlations between satelliteretrievals ofAOD,and cloud and precipitation properties. AI, defined as the product of AOD and the Angström exponent, which is reported as a better proxy than AOD to quantify CCN concentration due to its ability to weight AOD measurements towards the fine mode (Nakajima et al., 2001). Such correlationsBut Moderate Resolution Imaging Spectroradiometer (MODIS) retrieved are susceptible to various uncertainties arising from cloud contamination and the dependence of AOD on certain atmospheric components like water vapor(e.g.,Li et al., 2009;Boucher and Quaas, 2012).Moreover, because AOD isare limited as only measurable under cloud-free conditions, (Li et al.2009)causing a very low, the availability of AOD of AOD from Moderate Resolution Imaging Spectroradiometer (MODIS) AOD productsare available foris less than 30% of the timeover the PRD region (Wang et al., 2015). Therefore we cannot get enough AOD measurements, let alone the AI (Angström exponent is restricted over oceans because of its large uncertainties over land, Levy et al., 2010).All of these indicate huge uncertainty and severe limitations in using AOD here.. Very large uncertainties arise when usingAOD as a proxy for CCN (Andreae, 2009).These uncertainties can be reduced by applying the method proposed by Liu and Li(2014). However, the most serious problem inusing AOD as a proxy for CCN lies in the fact that AOD is only measurable under cloud-free conditionsand is subject to various retrieval errors, as critically reviewed byLi et al.(2009).

The ability of a particle to nucleate a cloud droplet dependson its size and its chemicalcomposition. The aerosol index (AI) is defined as the product of AOD and the Angström exponent, and is a good proxy to use to quantify CCN due to its ability to weight AOD measurements towards the fine mode(Nakajima et al., 2001; Andreae, 2009). A limitation of using the aerosol indexAI is that retrievals are restricted to over oceans because of the large uncertainties in Angström exponent retrievals over land(Levy et al.,2010). Furthermore, aerosols with an aerodynamic diameter less than 2.5 ($PM_{2.5}$) µm even 1 µm ($PM_1$) are also good proxies, (Seinfeld and Pandis, 1998) but with very limited observations.

Given the above problems,

Based on such limitations, in this study, we choose the ground-based $PM_{10}$ concentrations as a proxy of aerosol loading over the PRD region, which are available from 1 January 2007 to 31 December 2012. Vertical profiles of aerosols and clouds over the PRD region obtained from the Cloud-Aerosol Lidar and Infrared Pathfinder Satellite Observations mission show that aerosol particles are generally well-mixed in the boundary layer (Wang et al., 2015). Also, according to Anderson et al. (2003), the variability in aerosol properties at degrees spatial scale will not be very large. Therefore, $PM_{10}$ data should be good enough to indicate major aerosol episodes over the PRD region (~200km x 200km).

| Page 7: [96] Deleted | Jianping Guo | 8/8/18 7:19:00 AM |
|---|---|---|

Note that in order to exclude abnormal measurements, $PM_{10}$ is limited to ≤200µg/m³ in our study.

| Page 8: [97] Deleted | Jianping Guo | 8/9/18 10:11:00 PM |
|---|---|---|

The relationship between aerosols and precipitation structure can be established when the dataset is sorted out according to meteorological variables (Koren et al., 2012).

Due to the potential co-variations and influence of meteorological factors conditions on aerosol-precipitation interactionsinfluencing simultaneously aerosol concentration and precipitation, further investigation based on similar meteorological conditions it will be more feasible if the investigation of the co-variation of aerosol and precipitation is considered underis asked for single out the aerosol effect the samesimilar meteorological conditions on precipitation  based on ECMWF ERA-Interim reanalysis data(Uppala et al., 2008Koren et al., 2010). According to

previous studies, mMeteorological factors including the vertical pressure velocity (ω, Koren et al., 2012), wind shear between 850hPa (~1.5 km) and 500hPa (~5.5 km) (Fan et al., 2009), moisture flux divergence (MFD) from 1000hPa (near surface) to 400hPa (~7 km) (Khain et al., 2008), and convective available potential energy (CAPE, Dai et al., 1999) of surface layer parcel are further analyzed. ECMWF ERA-Interim reanalysis dataset provide parameters of ω, specific humidity (q), 'u' and 'v' component of wind (U and V) parameters used in this study include vertical velocity (ω), specific humidity, the "u" component of wind (U), the "v" component of wind (V), and convective available potential energy (CAPE) from ECMWF ERA-Interim reanalysis data.These dataare available four times a day, with a horizontal resolution of 0.125°×0.125° at pressure levels equal to 1000, 975, 950, 925, 900, 875, 850, 825, 800, 775, 750, 700, 650, 600, 550, 500, 450, and 400 hPa, and surface CAPE . with a horizontal/temporal resolution of 0.125°×0.125°/6-hourly.

| Page 8: [98] Deleted | huan | 7/24/18 7:21:00 PM |
|---|---|---|

The relationship between aerosols and precipitation structure can be establishedwhenthe dataset is sorted out according to meteorological variables(Koren et al., 2012).

| Page 9: [99] Deleted | huan | 8/7/18 9:01:00 AM |
|---|---|---|

dirtiest

| Page 9: [99] Deleted | huan | 8/7/18 9:01:00 AM |
|---|---|---|

dirtiest

| Page 9: [99] Deleted | huan | 8/7/18 9:01:00 AM |
|---|---|---|

dirtiest

| Page 9: [100] Deleted | Jianping Guo | 8/10/18 8:22:00 AM |
|---|---|---|

binbin

| Page 9: [100] Deleted | Jianping Guo | 8/10/18 8:22:00 AM |
|---|---|---|

binbin

| Page 9: [101] Deleted | huan | 8/7/18 9:03:00 AM |
|---|---|---|

range of $PM_{10}$ values defined for in

**Page 9: [102] Deleted**  Liu, Huan  7/3/18 9:17:00 AM

Data are divided into three groups to make sure that the daily mean $PM_{10}$ concentration exceeds the national air quality standard for the polluted case (75 μg/m3

**Page 9: [103] Deleted**  huan  7/28/18 9:16:00 AM

The first (lowest) bin represents relatively clean conditions and the third (highest) bin represents relatively polluted conditions.

**Page 9: [104] Deleted**  Liu, Huan  7/3/18 9:42:00 AM

$μg/m^3$,

| Page 9: [104] Deleted | Liu, Huan | 7/3/18 9:42:00 AM |

$\mu g/m^3$,

| Page 9: [105] Deleted | huan | 7/17/18 12:33:00 PM |

$\mu g/m^3$,

| Page 9: [105] Deleted | huan | 7/17/18 12:33:00 PM |

$\mu g/m^3$,

| Page 9: [106] Formatted | Jianping Guo | 8/11/18 11:29:00 PM |

fontstyle01, Font: Not Bold, Not Italic, (Asian) Japanese

| Page 9: [107] Deleted | huan | 8/7/18 9:11:00 AM |

On average, clean conditions for all precipitation types are defined when the daily mean $PM_{10}$ is <38$\mu g/m^3$ andpolluted conditions are defined when the daily mean $PM_{10}$ is >102 $\mu g/m^3$ (Table 2).

It alsocreates a sufficient contrast between clean and polluted groups while retaining good sampling statistics (Koren et al., 2012). Table 2 also summarizes the total number of profiles and the frequency of occurrence profiles(in %, relative to the total number of profiles) of profiles in the clean and polluted categories for each precipitation type.

To further examine aerosol influences on convective precipitation, this precipitation type is divided into three groups based on hourly R: light (R < 10 mm/h), moderate (10≤ R < 20 mm/h), and heavy (R ≥ 20 mm/h)

| Page 9: [107] Deleted | huan | 8/7/18 9:11:00 AM |

On average, clean conditions for all precipitation types are defined when the daily mean $PM_{10}$ is <38$\mu g/m^3$ andpolluted conditions are defined when the daily mean $PM_{10}$ is >102 $\mu g/m^3$ (Table 2).

It alsocreates a sufficient contrast between clean and polluted groups while retaining good sampling statistics (Koren et al., 2012). Table 2 also summarizes the total number of profiles and the frequency of occurrence profiles(in %, relative to the total number of profiles) of profiles in the clean and polluted categories for each precipitation type.

To further examine aerosol influences on convective precipitation, this precipitation type is divided into three groups based on hourly R: light (R < 10 mm/h), moderate (10≤ R < 20 mm/h), and heavy (R ≥ 20 mm/h)

**2.3.2 Meteorological variables**

Previous aerosol-precipitation interaction studies have suggested that atmospheric dynamic conditions and moisture fluxes areamong the most important meteorological variables contributing to changes in cloud properties and associatedprecipitation(Koren et al., 2010;Medeiros and Stevens, 2011;Jiang et al., 2011). To better isolate the aerosol effect, we need to determine the relative contributions of the following four meteorological factors to the variability in precipitation: ω, CAPE, vertical wind shear between 1000 850hPa (~1.5 km) and 700 500hPa (~5.5 km), and vertically integrated moisture flux divergence (MFD) from 1000 hPa (near surface) to 400hPa (~7 km).

CAPE is a measured for the amount of moist static energy for initiation of convection, and acts as an effective indicator of atmospheric instability, which has beenshown to be closely associated with the initiation ofprecipitation (Dai et al., 1999). For a fixedatmosphericcondition, wind shear can dictate whether aerosols suppress or enhance convective strength, depending on the atmospheric moisture and stability (Fan et al., 2009). MFD,another major factor in the formation of precipitation, determines the complex spatial variability of precipitation through the transport of water vapor(Khain et al., 2008).

The definition of MFDin units of g/(cm2s) is:

$$MFD = \nabla_P \cdot \left(\frac{\overrightarrow{V_H}q}{g}\right) = \frac{\partial}{\partial x}\left(\frac{\overrightarrow{V_H}q}{g}\right) + \frac{\partial}{\partial y}\left(\frac{\overrightarrow{V_H}q}{g}\right) \qquad (1)$$

$$\overrightarrow{V_H} = \vec{U} + \vec{V} \qquad (2)$$

Where$\overrightarrow{V_H}$represents the horizontal wind vector, $\vec{U}$and$\vec{V}$ represent the U and V components of wind inunits of m/s, q represents specific humidity in units of g/kg, P represents pressure in units of hPa, and g represents the acceleration due to gravity. MFD was calculated at 18 standard pressure levels:

1000, 975, 950, 925, 900, 875, 850, 825, 800, 775, 750, 700, 650, 600, 550, 500, 450, and 400hPa.

A negative MFD means convergence of water vapor and a positive MFD indicates divergence of

water vapor.

| Page 9: [109] Deleted | huan | 7/28/18 9:33:00 AM |
|---|---|---|

.

| Page 9: [110] Deleted | Jianping Guo | 8/12/18 4:28:00 PM |
|---|---|---|

normalized

| Page 9: [111] Deleted | huan | 7/28/18 9:25:00 AM |
|---|---|---|

NCFAD.

| Page 9: [112] Deleted | huan | 7/28/18 9:26:00 AM |
|---|---|---|

as observed by PR

| Page 9: [113] Deleted | Jianping Guo | 8/12/18 4:28:00 PM |

contoured frequency by altitude

| Page 9: [114] Formatted | Jianping Guo | 8/11/18 11:29:00 PM |

Superscript

| Page 9: [115] Deleted | huan | 8/7/18 9:16:00 AM |

'

| Page 10: [116] Deleted | huan | 8/3/18 1:38:00 PM |

.

| Page 10: [117] Formatted | Jianping Guo | 8/11/18 11:29:00 PM |

Font: +Body (Cambria), Check spelling and grammar

| Page 10: [118] Deleted | huan | 7/28/18 9:36:00 AM |

.

| Page 10: [118] Deleted | huan | 7/28/18 9:36:00 AM |

.

| Page 10: [118] Deleted | huan | 7/28/18 9:36:00 AM |

.

| Page 10: [118] Deleted | huan | 7/28/18 9:36:00 AM |

.

| Page 10: [118] Deleted | huan | 7/28/18 9:36:00 AM |

.

| Page 10: [119] Deleted | Jianping Guo | 8/10/18 3:00:00 PM |

**2.3.4 Discrimination between synoptic-scale and localized rain events**

Generally speaking, the main contribution of rainfall is synoptic-scale precipitating systems that characterized by horizontal length scales of the order of 1000 km or more, such as frontal passages or low-pressure systems. However, such synoptic-scale precipitating systems are normally firstly driven by pressure gradient term, and the local atmospheric condition just play very limited role. Therefore, in order to single out precipitating systems that affect by local atmospheric conditions much directly and ensure the representativeness of ground-based $PM_{10}$ concentration, we focus on localized precipitating systems which are characterized by thermal-driven convective clouds only (Guo et al., 2017; Lin et al., 2018).

As we mentioned, synoptic-scale precipitating systems of large horizontal length scales, while localized precipitating systems should be small and random. Therefore, we can easily identified the synoptic-scale precipitating systems on weather maps, while others can be refer to localized precipitating systems. All the day with both ground-based aerosol observations and TRMM precipitation measurements are checked based on weather charts manually. First, for each day with valid precipitation (>0.1mm/day) over PRD, the daily averaged wind field at 850hPa (~1.5 km) pressure level was plotted along with geo-potential height at 500hPa (~5.5 km) pressure level. Then, such weather maps are analyzed manually to decide if we can figure out any weather patterns

that favor the onset and development of synoptic-scale precipitation or not. If yes, precipitation events during this day are classified as synoptic-scale rain events; if not, precipitation events during this day are classified as localized rain events. As such, the total number of collocated localized rain events reached up to 253[h8], whereas 194 for synoptic-scale rain events.

Figure 2 show typical weather maps for both synoptic-scale and localized rain events. On 26 June 2008 (Figure 2(a)), PRD lies at the bottom of the weak low pressure at 500hPa pressure level. At 850hPa pressure level, there is a weak cyclone on the left-forward side of PRD, where a south-western to north-eastern low-level jet stream overpasses at the same time, leading to strong water vapors adverted accumulation over PRD from South China Sea. More importantly, the wind shear observed at 850hPa pressure level is most favorable for the formation and evolution of precipitation. Therefore, rain events observed under such weather patterns can be thought as synoptic-scale rain events. In contrast, as shown in Figure 2(b), PRD is largely controlled by the subtropical high-pressure areas, in combination with the anti-cyclone systems at low levels on 2 July 2008, which will be generally thought as no weather patterns that favor for large-scale convections. Therefore, rain events during this day can be attributed to localized thermal convections with high certainty, and are identified as localized rain events for further analysis.

| Page 10: [120] Formatted | Jianping Guo | 8/11/18 11:29:00 PM |
|---|---|---|

Subscript

| Page 10: [120] Formatted | Jianping Guo | 8/11/18 11:29:00 PM |
|---|---|---|

Subscript

| Page 10: [120] Formatted | Jianping Guo | 8/11/18 11:29:00 PM |
|---|---|---|

Subscript

| Page 10: [121] Deleted | huan | 7/28/18 9:46:00 AM |
|---|---|---|

where $Z$ is the measured radar reflectivity in dBZ, $H$ is the height above ground in km, and $i$ is an index from 1 to 80, representing different levels in the atmosphere. Alarger magnitude valueof ZCOG means that the precipitation system has developed to a higher level in the atmosphere, indicating stronger convection.

| Page 10: [122] Deleted | huan | 7/24/18 7:05:00 PM |
|---|---|---|

3.1 Regional aerosol features

PM$_{2.5}$ began to be measured as of 2013,largely due to the "January 2013" severe haze event shrouded over the whole eastern China. China central government decided to make great efforts in attempt to address the increasingly serious air quality issues across the board, including setting up the PM$_{2.5}$ criteria, among others. Therefore, PM$_{2.5}$ measurements during the period of January 2007 - December 2012 do not exist. It is still an efficient alternative way to use the yearly averaged PM data during the period of November 2013-October 2014 to characterize the regional aerosol features in the PRD region.Figure 1a presents the spatial distribution of meanPM$_{10}$concentrationscollected in the PRD regionfrom November 2013 to October 2014.Nearly 60% of the measurement sites are characterized by high PM$_{10}$ concentrations (>70 μg/m$^3$). This value (70 μg/m$^3$)is the World Health Organization (WHO) interim target 1 annual mean level, which is associated with about a 15% higher long-term mortality risk relative to the WHO air quality guideline level of 20 μg/m$^3$(WHO, 2006).

Figure 1b shows the ratio of annual mean PM$_{2.5}$to annualmean PM$_{10}$. Most megacities (e.g., Guangzhou and Shengzhen) are characterized by a high ratio of PM$_{2.5}$ to PM$_{10}$(> 0.7). This suggests that fine PM, which is mostly generated by anthropogenic activities such as daily power generation and industrial production, dominates aerosol pollution in this area. This region isan ideal testbedto probe the aerosol impact on 3D precipitation structures.

3.2 Discrimination betweensynoptic--scale and local-scaleized precipitating systems

Generally speaking, synoptic--scale precipitation involves frontal passages or low-pressure systems (weather patterns helping the onset and development of large-scale convection), as compared with local-scaleized precipitation characterized by thermal-drivenconvective clouds (weather patterns show no helping conditions to large-scale convection), which is relatively small and random and fed by the boundary layer air. Therefore the locally aerosol particles can affect these localized events much directly (aerosol). Our recent study(Guo et al., 2017) indicates that local-scale precipitation events are more closely linked to aerosol relative to synoptic-scale precipitation. In order to make sure that only precipitating system more susceptible to the local boundary layer aerosol were considered, all the satellite scenes with synoptic precipitation were excluded.For any given day,ground-based aerosol observations have tocollocate with precipitation measurements from TRMM in attempt to obtain a valid data pair.As such,the total number of

collocated samples reached up to 255 for local-scaleized precipitation events, whereas 194 for synoptic-scale precipitation events.

The local-scale precipitation eventwas determined based on the weather charts, wheredaily averaged wind field at 850 hPawas plotted along withgeo-potential height at 500hPa. Note that the discrimination was manually performed through visual interpretation of the weather plot for each day with valid precipitation (>0.1mm/day) over PRD, owing to the extreme complexities in discriminating the weather systems for local- izedand synoptic-scale precipitations.To make it clear, subplots in Fig 2 show typical weather chartsfor both synoptic-scale and localized precipitation events

Figure 2 illustrates two typical weather plots, corresponding to .synoptic- and local-scale precipitation events, respectively. On 26 June 2008 (Figure 2a),PRD lies at the bottom of the weak low pressure at 500 hPa level. At 850 hPa level, there is a weak cyclone on the left-forward side of PRD, wherea south-western to north-eastern low-level jet stream overpasses at the same time, leading to strong water vapors advectedadvertedover PRD from South China Sea. More importantly, the wind shear observed at 850 hPa is most favorable for the formation and evolution of precipitation. OverallTherefore,precipitationevents observed under the suchweather patterns at both 500 hPa and 850 hPahelp the onset and development of large-scale convection, sothis precipitation event occurred over PRDweather patterns can be thoughtofas atypical synoptic-scale precipitation event. In contrast, as shown in Figure 2b, PRD is largely controlled by the subtropical high-pressure areas, in combination with the anti-cyclone systems at low levels on 2 July 2008, which will be generally thought as no synoptic systems.,as shown in Figure 2b. Therefore, this precipitation can be attributed to local thermal convection with high certainty.

| Page 10: [123] Deleted | Jianping Guo | 8/10/18 9:29:00 PM |

**3.1 Regional aerosol features**

[revised manuscript text omitted]

Page 35: [153] Deleted huan 8/5/18 10:14:00 AM
18 dBz radar echo

heightdifferences(

Page 35: [153] Deleted          huan          8/5/18 10:14:00 AM
18 dBz radar echo heightdifferences(

Page 35: [153] Deleted          huan          8/5/18 10:14:00 AM
18 dBz radar echo heightdifferences(

Page 35: [153] Deleted          huan          8/5/18 10:14:00 AM
18 dBz radar echo heightdifferences(

Page 35: [153] Deleted          huan          8/5/18 10:14:00 AM
18 dBz radar echo heightdifferences(

Page 35: [153] Deleted          huan          8/5/18 10:14:00 AM
18 dBz radar echo heightdifferences(

Page 35: [153] Deleted          huan          8/5/18 10:14:00 AM

18 dBz radar echo heightdifferences(

Page 35: [153] Deleted huan 8/5/18 10:14:00 AM

18 dBz radar echo heightdifferences(

Page 35: [153] Deleted huan 8/5/18 10:14:00 AM

18 dBz radar echo heightdifferences(

Page 35: [153] Deleted huan 8/5/18 10:14:00 AM

18 dBz radar echo heightdifferences(

Page 35: [153] Deleted huan 8/5/18 10:14:00 AM

18 dBz radar echo heightdifferences(

Page 35: [153] Deleted huan 8/5/18 10:14:00 AM

18 dBz radar echo heightdifferences(

Page 35: [153] Deleted huan 8/5/18 10:14:00 AM

18 dBz radar echo heightdifferences(

Page 35: [153] Deleted huan 8/5/18 10:14:00 AM

18 dBz radar echo heightdifferences(

Page 35: [153] Deleted huan 8/5/18 10:14:00 AM

18 dBz radar echo heightdifferences(

Page 35: [153] Deleted huan 8/5/18 10:14:00 AM

18 dBz radar echo heightdifferences(

| Page 35: [153] Deleted | huan | 8/5/18 10:14:00 AM |
|---|---|---|

18 dBz radar echo heightdifferences(